# Implementation of an Artificial Intelligence Approach to GPR Systems for Landmine Detection

**Oleksandr A. Pryshchenko** [1], **Vadym Plakhtii** [1], **Oleksandr M. Dumin** [1], **Gennadiy P. Pochanin** [2], **Vadym P. Ruban** [2], **Lorenzo Capineri** [3] **and Fronefield Crawford** [4,*]

1   Department of Applied Electrodynamics, V. N. Karazin Kharkiv National University, 61022 Kharkiv, Ukraine
2   Department of Radiophysical Introscopy, O. Ya. Usikov Institute for Radiophysics and Electronics of the National Academy of Science of Ukraine, 61085 Kharkiv, Ukraine
3   Department of Information Engineering, Università degli Studi di Firenze, Via S. Marta 3, 50139 Florence, Italy
4   Franklin & Marshall College, P.O. Box 3003, Lancaster, PA 17604, USA
*   Correspondence: fcrawfor@fandm.edu

**Abstract:** Artificial Neural Network (ANN) approaches are applied to detect and determine the object class using a special set of the UltraWideBand (UWB) pulse Ground Penetrating Radar (GPR) sounding results. It used the results of GPR sounding with the antenna system, consisting of one radiator and four receiving antennas located around the transmitting antenna. The presence of four receiving antennas and, accordingly, the signals received from four spatially separated positions of the antennas provide a collection of signals received after reflection from an object at different angles and, due to this, to determine the location of the object in a coordinate system, connected to the antenna. We considered the sums and differences of signals received by two of the four antennas in six possible combinations: (1 and 2, 1 and 3, 2 and 3, 1 and 4, etc.). These combinations were then stacked sequentially one by one into one long signal. Synthetic signals constructed in such a way contain many more notable differences and specific information about the class to which the object belongs as well as the location of the searched object compared to the signals obtained by an antenna system with just one radiating and one receiving antenna. It therefore increases the accuracy in determining the object's coordinates and its classification. The pulse radiation, propagation, and scattering are numerically simulated by the finite difference time domain (FDTD) method. Results from the experiment on mine detection are used to examine ANN too. The set of signals from different objects having different distances from the GPR was used as a training and testing dataset for ANN. The training aims to recognize and classify the detected object as a landmine or other object and to determine its location. The influence of Gaussian noise added to the signals on noise immunity of ANN was investigated. The recognition results obtained by using an ANN ensemble are presented. The ensemble consists of fully connected and recurrent neural networks, gated recurrent units, and a long-short term memory network. The results of the recognition by all ANNs are processed by a meta network to provide a better quality of underground object classification.

**Keywords:** impulse subsurface radars; GPR-multichannel system; digital signal processing; landmine detection; shallow GPR survey; artificial neural network ensembles

## 1. Introduction

Explosives of various kinds contaminate the territory of the Donetsk and Luhansk regions as a result of military conflicts in the eastern part of Ukraine [1]. Among the most hazardous of these objects are antipersonnel landmines. The demining of such areas is commonly carried out by military sappers, but the concept of creating remotely controlled devices for this purpose is of great interest [2–4]. During the last decade, techniques have been developed to combine sensors of different types and fuse their data to achieve more reliable mine detection and to decrease the false alarm rate [2,5,6].

One promising approach to this uses UWB GPR [7], which radiates ultrashort electromagnetic impulses to provide high spatial resolution of a survey region [8]. There is a wide range of applications for this technology, such as detecting humans, including people hidden behind opaque obstacles [9], soil analysis for subway construction [10], and the humanitarian demining activities mentioned above [11]. Mounting GPR systems on robotic platforms [2,12] and on drones [13] provides additional safety.

One significant issue with any device working in a mined area is the false positive rate of mine recognition. This results from the presence other kinds of buried objects in the vicinity besides mines.

Successful implementation of GPR techniques requires special signal processing methods. Among these are the wavelet transform [14], a semi-analytic mode matching algorithm [15], the generalized Hough transform [16], and the correlation method [17,18]. As shown in practice, holographic radars are the most reliable devices to overcome false positives in mine detection. They are used to identify mines of different classes using signal processing [19,20].

In this article, we present an artificial intelligence approach to the problem, specifically using neural networks [21]. They are used for different applications [22–24], and their speed and effectiveness can be increased using neuroprocessors [25].

The effective use of artificial neural networks for automatic identification of defects of this type is proposed in [26]. To reduce the information load on the network and speed up the learning process, it is important to pre-process the images obtained during the experiment. Canny edge detection operator is good for this purpose. This makes it possible to highlight cracks and defects in the background of the entire image and converts the image into a binary format, which greatly facilitates the further operation of the artificial neural network. This approach has shown itself well when processing photos in different lighting quality, which characterizes its stable operation in real conditions.

An interesting and effective approach [27], which includes the use of artificial intelligence, has also been proposed for the tasks of visual assessment of the condition of infrastructure objects. Its peculiarity is the proposition to use the fusion features-based broad learning system. This made it possible to speed up the learning process compared to conventional deep neural networks. It is also important to note the possibility of scaling such a system, because the process of additional training has also been simplified.

For the task of predicting faults on gas pipelines, the authors have well-demonstrated the useful features of backpropagation neural network and support vector machines. As a result, in 850 cases, they were able to effectively analyze possible breakdowns and find key causes that led to malfunctions [28].

Neural networks require a training process before their use. This can be provided by using a training dataset, which can be quite difficult to obtain due to limited access to information, the high cost of obtaining the data, or the complexity of the computer calculations. In our case, the training dataset can be collected from the reflected impulse signals [29]. The more complex the object under investigation is, the more training data we require for identification [30]. Fortunately, many studies have shown the stability of ANN in working with noisy signals [31–34].

The application of neural network ensembles [35] can improve the effectiveness of the system in comparison with a single network realization. The collective voting for the best answer has already proven to be a valid approach to the problems of Hyperspectral Image Classification [36], co-reference resolution [37], Computer Vision and NLP [38], and healthcare [39]. Consequently, it is worth deploying this idea to address the mine detection problem.

The purposes of this work are to demonstrate the benefits of the ANN application for GPR signal analysis for landmine detection and positioning. Section 2 describes the design of the 1Tx + 4Rx antenna system, a special way to arrange signals for analysis with ANN, and models of objects chosen for investigation. Section 3 examined the influence of additive noise in received signals on the object classification quality. Here we used numerically

simulated and experimental data. At the end of Section 3, we studied what advantages could be obtained if we apply ensemble learning to identify and classify subsurface objects. Finally, conclusions are formulated in Section 4.

## 2. Statement of the Problem

### 2.1. Description of the GPR and the Working Pipeline

Consider the problem of detecting a subsurface object using UWB GPR with a 1Tx + 4Rx antenna system (AS) (Figure 1). The AS is located at the height Zo = 32 cm above the surface of the ground with a relative permittivity $\varepsilon$ = 9 and conductivity $\sigma$ = 0.005 S/m. The subsurface object is 'buried' at the depth 1 cm under the ground surface in the location with coordinates Xo and Yo in the coordinate system X, Y, and Z associated with the antenna system. At GPR sounding, the AS moves along the *Y* axis (continuously in experiments or by steps in calculations) and collects radar data. The Tx antenna is excited by a Gaussian pulse of 0.23 ns duration [34]. It means that the spectrum of the voltage impulse exciting the transmitting antenna covers the frequency range from 0 to 4.4 GHz. Four receiving antennas (4Rx) receive electromagnetic waves coming to them as combinations of direct coupling waves, reflections from other antennas, reflections from the ground surface (shown in Figure 1), and reflections from the subsurface object (shown in Figure 1).

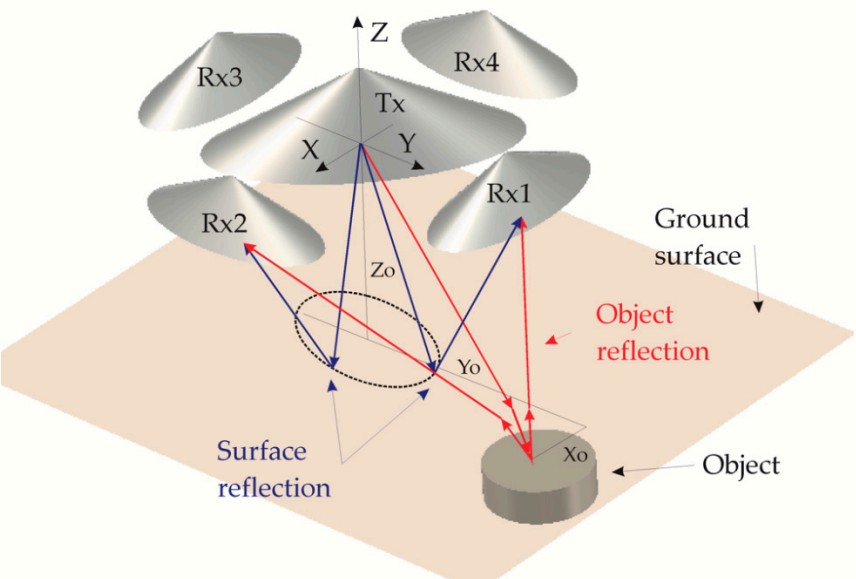

**Figure 1.** Schematic illustration of the subsurface sounding by the GPR with the 1Tx + 4Rx antenna system. GPR pulse injection from Tx. Reflected waves: blue corresponds to surface reflection; red corresponds to object reflection. Waves are shown only for two (Rx1 and Rx2) receiving antennas. Antenna system is located at the height Zo = 32 cm over the ground surface.

Reflections for Rx1 and Rx2 are hidden in Figure 1.

Four Rx are located around the Tx antenna at the same distance (10 cm) from it with an angular spacing of 90°. All Rx antennas were oriented in such a way that their polarizations were oriented at angles ±45° to the polarization of the Tx antenna (Figure 2b).

The same antenna design was used in the UWB GPR for landmine detection [2,40]. To detect a subsurface object and determine its position, we used meanings of times of flight of sounding signals from the Tx antenna to the object and back to all four Rx antennas.

In this paper we applied ANN for the detection, classification, and positioning of different subsurface objects.

The received signals require pre-processing to work with the ANN. The pipeline of the detection system (Appendix A) consists of signal resampling with equal for all signals time steps 10 ps ($10^{-11}$ s), normalization using the square root of the signal energy, and

forming a single signal containing the resampled signals. Such pre-processing is performed for the training and testing data for the ANN. Additionally, resampling allows the data from different experiments and simulations to be used as signal sets for the training of the ANN.

Top view                                                                 Bottom view

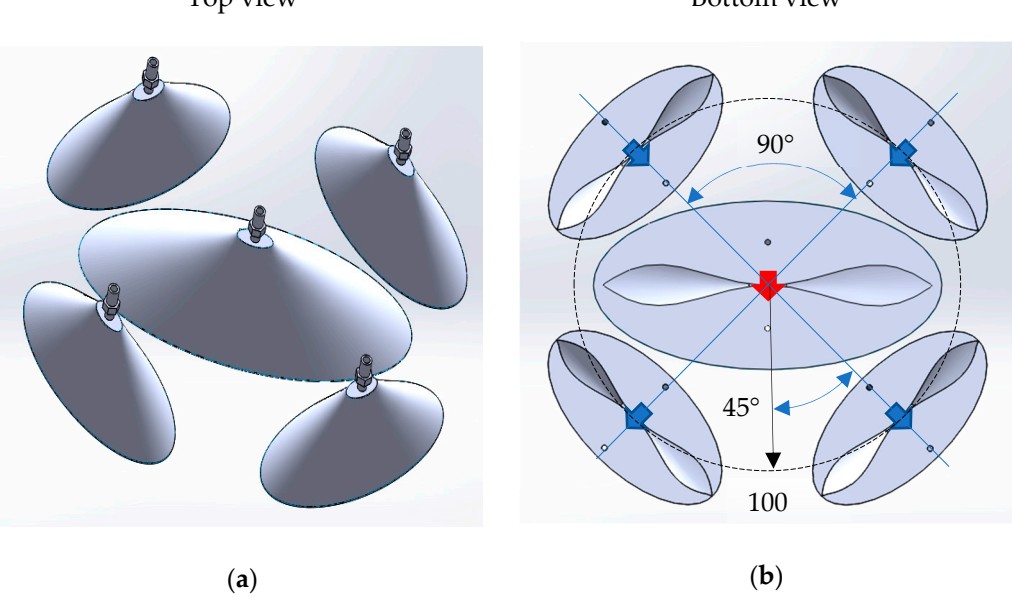

(**a**)                                                                 (**b**)

**Figure 2.** Model of the GPR antenna system (**a**). The four receiving antennas (blue arrows denote Rx antennas ports) are distributed in the corners of a square around the transmitting antenna (red arrow denotes Tx antenna port) (**b**). Distances between the center of the transmitting antenna and the centers of receiving antennas are 100 mm. The polarizations of receiving antennas are oriented at angles ± 45° to the polarization of the transmitting antenna.

The set of four Rx simulated signals for sounding at one of the steps is shown in Figure 3. As we can see, a significant part of signals corresponding to the direct coupling and reflection from the surface has the same waveform (differs only in polarity). At the same time, for better recognition difference between signals should be more notable.

To mitigate the influence of the direct coupling between Tx and Rx antennas and reflections between all antennas, we applied a background removal procedure. However, in contrast to the commonly used approach in which the averaged signal is calculated on the base of all received signals and then subtracted from the sounding results, or the signal is measured at the place where the object is definitely absent and then subtracted from the results of sounding, in our case we have four received direct coupling signals at the single place of measurement. Owing to the symmetry of the antenna system, the direct coupling signals and reflections from the ground surface are identical waveforms, and it allows removing background and reflection from the ground surface using simple math procedures (adding or subtracting). Figure 4 shows six combinations of initial signals $S1 - S2/S1 + S3/S1 + S4/S2 + S3/S2 + S4/S3 - S4$, providing mitigation of clutter conditioned by both the direct coupling and reflected by the ground surface signals. It forms a unique set of stitched-in sequence non-repeated signals for future processing with ANN.

Such an approach is slightly different from preparing real GPR A-scans, including the algorithm described in [41]. The proposed algorithm does not require the additional measurements of waves reflected from the ground without a hidden object or averaged signals for subtraction. The advantages of this approach also consist in the sufficiency of only four received signals to eliminate the clutter. There is no necessity to wait while all sounding data at the whole path will be recorded and averaged for subtracting. There is no doubt whether the subsurface object exists at the starting point.

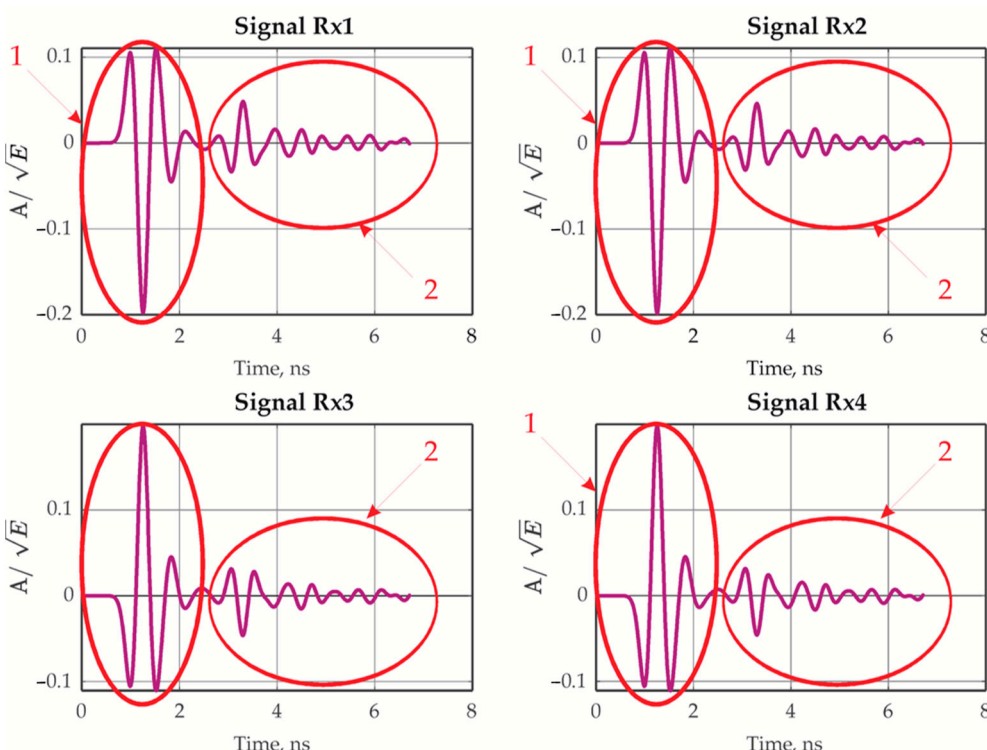

**Figure 3.** Example of signals received by the 4Rx antenna system of the GPR, where region 1 is the signal of direct coupling between the Tx and Rx antennas and the signals reflected by other antennas of the system, and region 2 is the main reflection window time covering reflection from the ground surface and subsurface object (if it present in the ground).

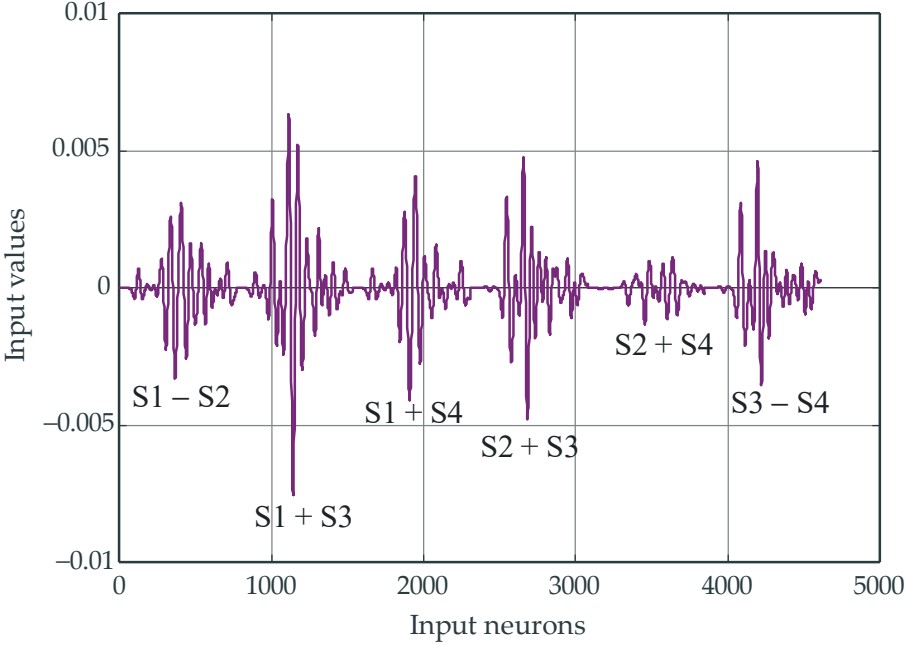

**Figure 4.** Example of training/testing data samples for the neural network which consists of 6 stitched pre-processed signals.

In the proposed way, we can exclude from consideration the powerful direct coupling signal and, most importantly, increase the informative low-amplitude components of the received signal.

For the future processing of the stitched in sequence signals (Figure 4), we used a fully connected ANN (Figure A2) containing seven layers. The first (input) layer contains 4614 neurons, corresponding to the dimension of the array of the stitched data (six combinations of signals containing 769 samples recorded with time steps of 10 ps as demonstrated in Figure 4). The subsequent five hidden layers have 4000 neurons in each one. Finally, the output layer has 65 neurons, corresponding to eight objects of interest at eight possible distances from the antenna system for each object and additional output indicating the presence or absence of an object in a given area. As an activation function, we have chosen a hyperbolic tangent.

To take into account all of the features of the GPR, the soil material parameters, and detailed constructions of underground object models, the electromagnetic problem is solved with the finite difference time domain (FDTD) method.

### 2.2. Description of Underground Objects

The antipersonnel mine types that are widely disseminated in the Donetsk and Luhansk regions are simulated as metal-dielectric objects in this work. The model of the PMN-1 mine has a height of 53 mm and a diameter of 110 mm (see Figure 5). The body material of the PMN-1 (orange) is bakelite, and its top (gray) is rubber. In addition, there is an intermediate volume with air, a metal element, and the explosive substance with electrical characteristics $\epsilon = 3.1$ and $\sigma = 0.0044$ S/m [42,43].

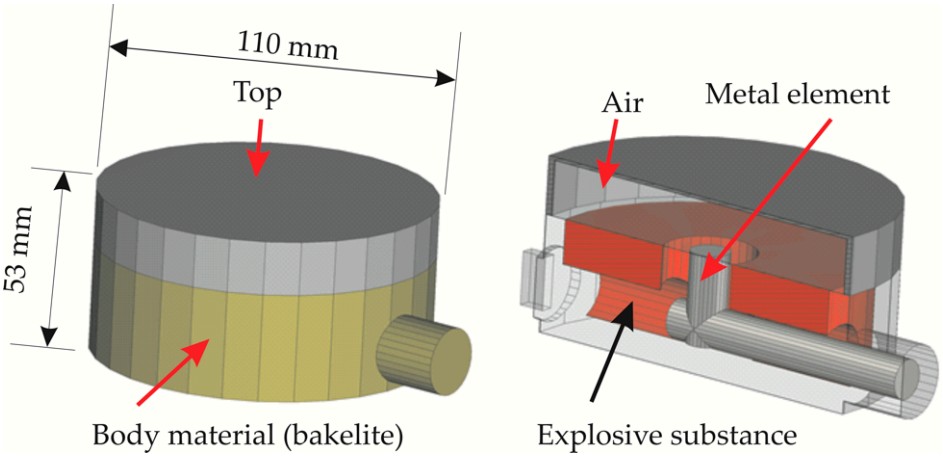

**Figure 5.** Schematic image of PMN-1 (on the **left**) and its cross section (on the **right**).

The model of the PMN-4 mine depicted in Figure 6 has slightly different dimensions and structure, but the body materials and electrical characteristics of the explosive are the same as for PMN-1. For the problem of humanitarian demining, the ability of an analyzing system to accurately classify an object by its reflected waveform is very important, especially for real-time identification [44]. We need to reliably identify what is buried immediately in front of the detection system, either a dangerous mine or harmless clutter (for example, used cans). Therefore, to approximate a real subsurface sounding situation, we included some artificial interfering objects, such as used metal tins, with different properties. The neural network is trained for the classification of these alternate objects as well as for the specified mine types.

The two most common types of cans in Ukraine were used for the numerical simulation of the problem. The first of these has a diameter of 10 cm and a height of 3.5 cm. Under the ground they can be presented in three possible states, named can1, can2, and can3, which are described and presented in Figure 7a–c. The second can under consideration has the same configuration, but has a different size: a diameter of 8.5 cm and height of 5 cm. They can be in the same three states as the first can, where can4 is an open can without a cap (such as can1), can5 is an open can with a cap attached (such as can2), and can6 is a hollow

tin with a closed cap (such as can3). With this, we enhanced the simulation, and the ANN can learn to distinguish an antipersonnel mine from different obstacles such as cans.

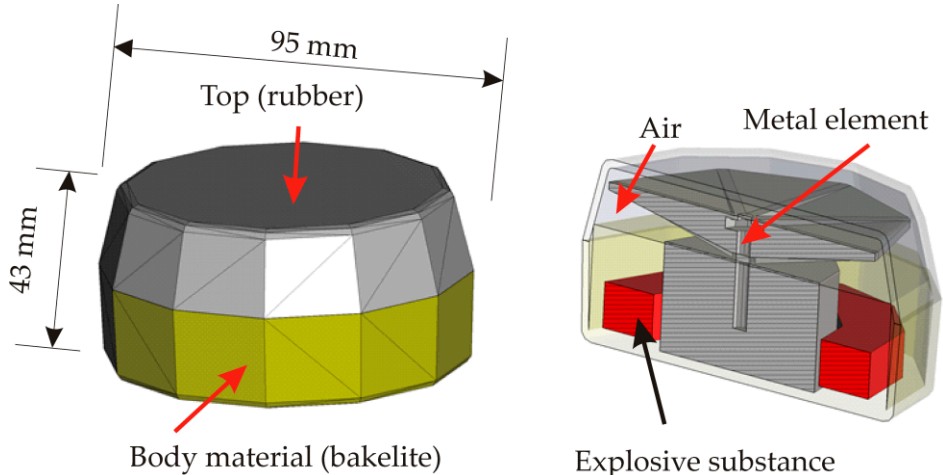

**Figure 6.** Schematic image of PMN-4 (on the **left**) and its cross section (on the **right**).

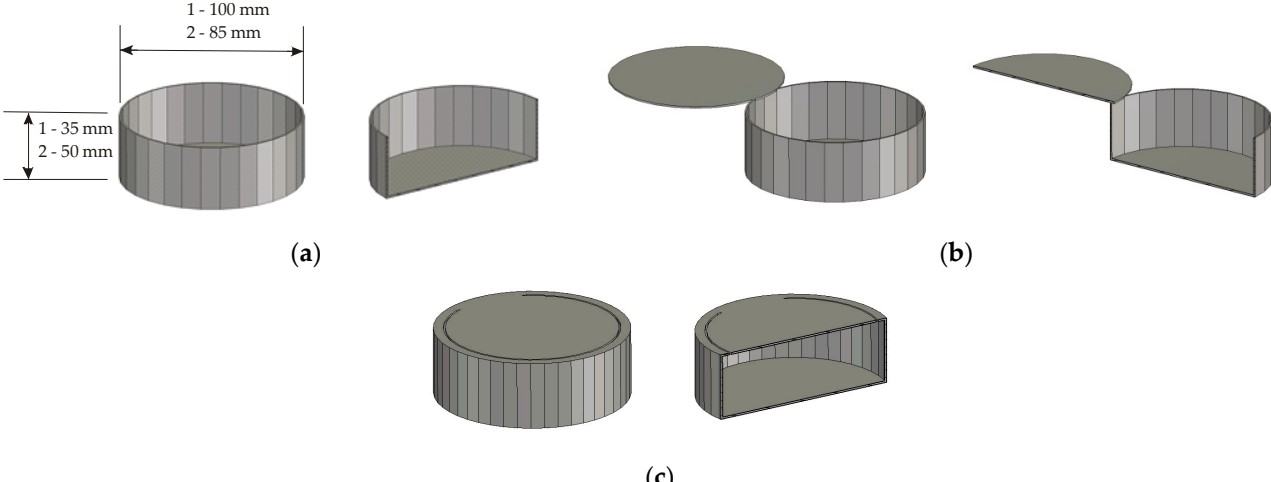

**Figure 7.** Three possible states of metal can and its cross-section, where can1 is an open can without a cap (**a**), can2 is an open can with a cap attached (**b**), and can3 is a can with a closed cap (**c**).

Future research will investigate how our approach works for the detection of mines having a larger dielectric material content. A suitable candidate for this investigation is the PFM mine, depicted in Figure 8, which is being used in the Donbas and Lugansk regions. It is 12 cm long, 2 cm tall and 4.6 cm in height. A comparison of this mine to the PMN-1 and PMN-4 mines shows that the PFM's detonation mechanism is much smaller. This leads to a smaller reflection of the electromagnetic impulse, which complicates detection. In addition, the permittivity of the other non-metallic components of the mine are quite close to the ground permittivity, making the PFM mine's body and explosive material almost invisible to the radar.

Another complicating factor in identifying the PFM mine is its non-symmetrical form, in contrast to the PMN-1 and PMN-4 mines and the metal tins, where the symmetry can be traced in at least one plane. This is an important factor since a totally different form of the reflected impulse will arrive from the different angles, which can make detection and identification more complicated. To provide better recognition we decided to calculate the PFM mine response in two orientations relative to the GPR. These are a cross orientation and a longitudinal orientation, as shown in Figure 9. This gives the neural network a more accurate description of the object, at least in two planes.

Moreover, in order to simulate the conditions of a more realistic subsurface survey, a white noise is added to the initial four signals but only after the background subtraction [41]. The noise sufficiently decreases the signal quality after the subtraction for the given level of signal-to-noise ratio (SNR) in the received signals. This occurs because the separation of the low-energy part of a signal containing useful information about an object significantly decreases the signal maximum, reducing the SNR.

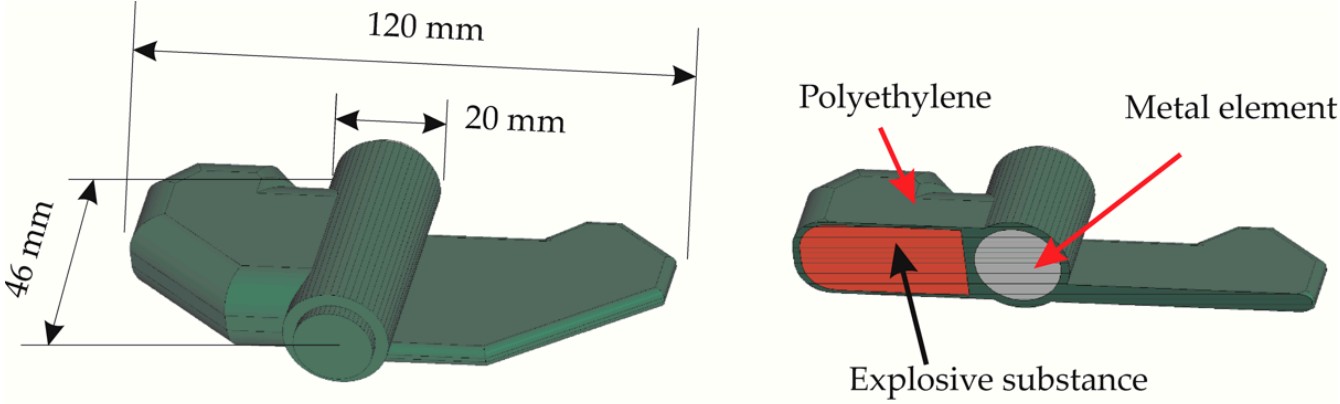

**Figure 8.** Schematic image of PFM (on the **left**) and its cross section (on the **right**).

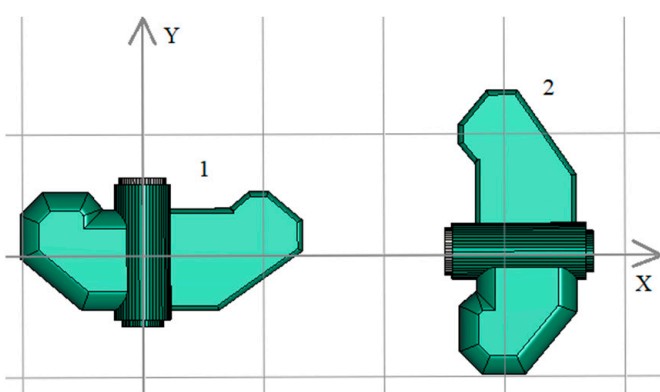

**Figure 9.** Two orientations of the PFM mine relative to the GPR. 1—cross orientation, 2—longitudinal orientation.

This effect is illustrated in Figure 10, where we compare the time shape of the signal received by one of the four antennas to that of a pre-processed signal. In addition, the normalization by the square root of its energy is applied. For a given SNR = 25 dB, the raw received signal only slightly changes its time form, while the pre-processed signal is seriously distorted by the Gaussian noise, as seen in the last graph in Figure 10 where SNR = −7.4 dB. Thus, it is difficult for the neural network to extract useful information from noisy data.

It should be noted that the result of the neural network recognition can significantly vary with high levels of additive Gaussian noise. These changes do not depend on the SNR level, but on the random realizations of the noise distribution in every data sample or on the parameters of the numerical simulations [12]. Therefore, to obtain a statistical generalization of ANN results we simulated 1000 random realizations of noise with a constant level of SNR for every testing data sample.

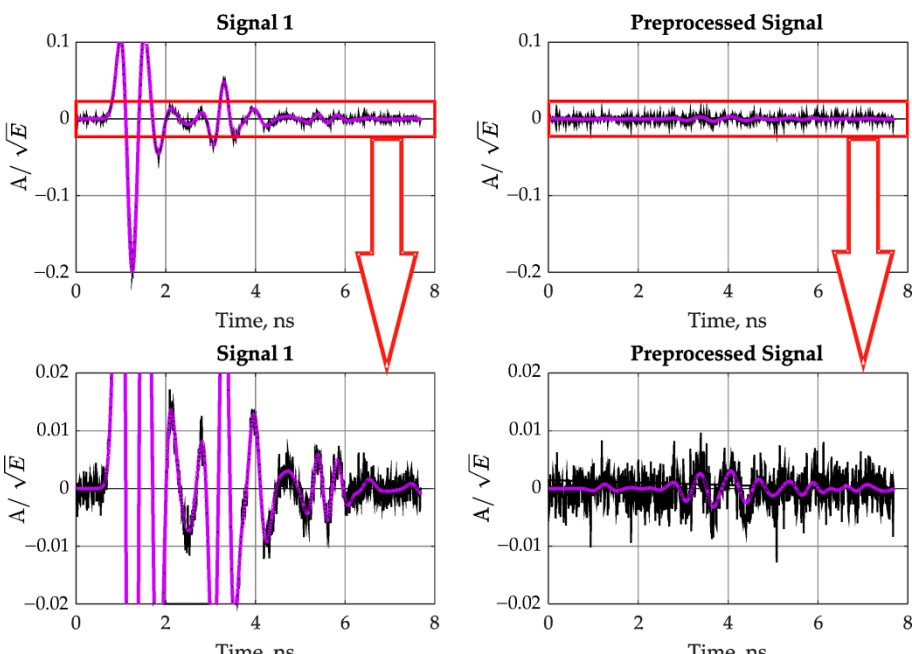

**Figure 10.** Illustration of different influences of the same SNR level (25 dB) in the raw and pre-processed signal cases.

## 3. Results and Discussion

### 3.1. Mine Detection in Simulated Data

Parameters of simulation for the Section 3.1: the sample interval is 10 ps; whole time window is 7.69 ns; at simulation and data processing the object was placed at Yo = 0; the sampling time is 10 ps. A whole time window is 7.69 ns; at simulation and data processing, the object was at Xo = 0 cm and Yo = 0 cm; 5 cm . . . 35 cm. The depth of the object under the surface of the ground is 1 cm.

The number of recognitions of each object for different distances of the reflected wave from the PMN-4 mine and for different SNR values is presented in Figure 11. It is seen that the ANN shows satisfactory noise immunity in detecting the PMN-4 mine at a distance of Yo = 20 cm (Figure 1). A statistically correct answer can be traced up to an SNR level of 5 dB, which is a very good if we take into account the conditions of experimental investigations.

The results from using the same positions but with a PMN-1 mine are shown in Figure 12, where we see that the results are less stable with the presence of noise. In this case the answer becomes unclear at an SNR level of 10 dB. However, this limitation can be treated as acceptable for real subsurface surveys.

We have also investigated the neural network performance on the border simulation positions of the objects, namely, when the object is under the center of the GPR antenna system (i.e., where the object's horizontal distance is 0 cm and where the object is at the maximum possible distance from the radar, which is 35 cm in height). We see that if a PMN-1 (Figure 13) or PMN-4 (Figure 14) mine is under the antenna system, then at high noise values the ANN will work better for PMN-1. However, in both cases we see stable classification results. This may be a consequence of training dataset enrichment with false objects (cans).

We considered the most distant possible positions of the investigated objects. The results of the recognition tests are presented in Figures 15 and 16. From these graphs we see a reliable classification for high SNR levels. The results for noise stability are similar, but the recognition of the mine is better in the PMN-4 case.

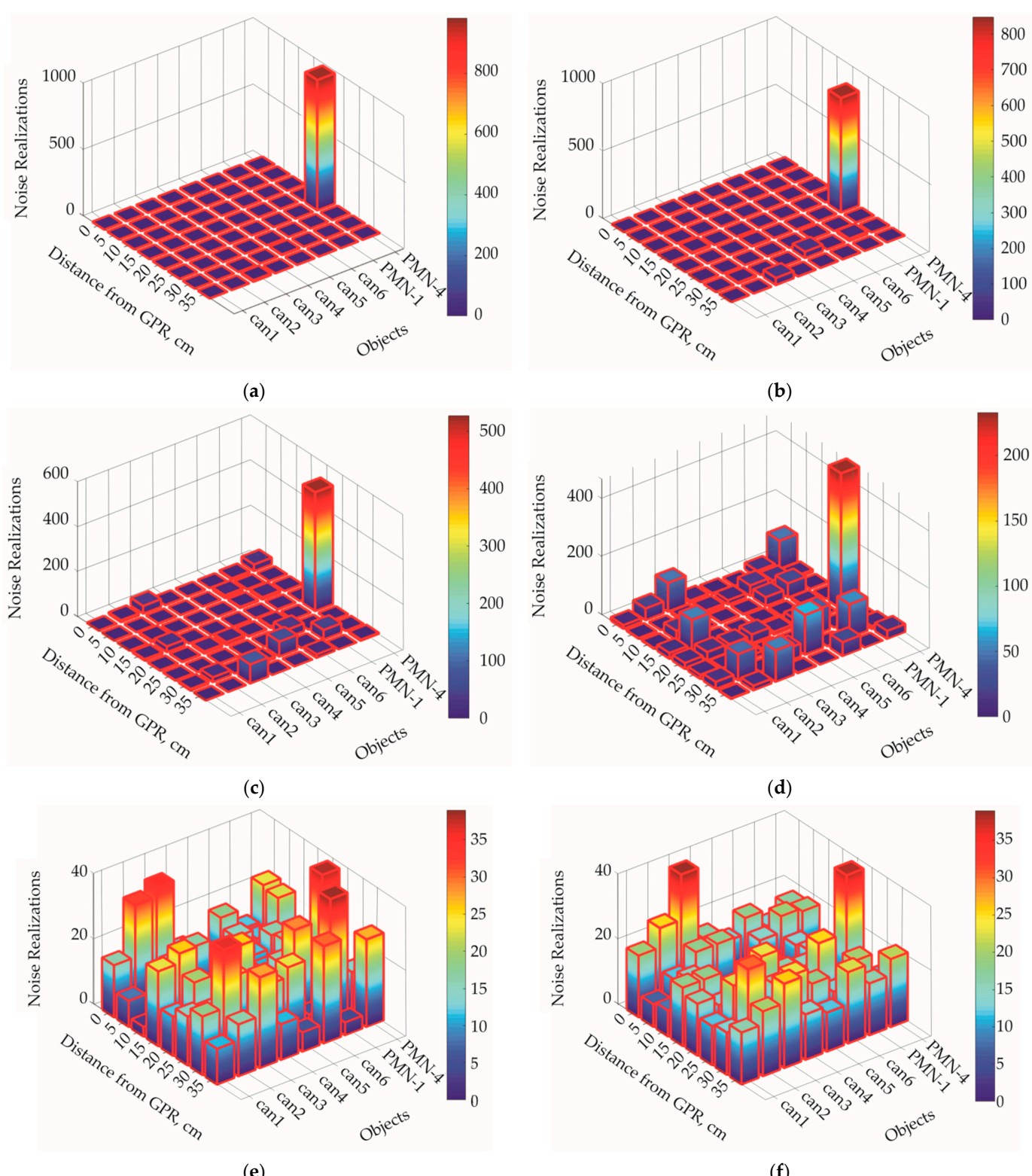

**Figure 11.** Neural network response statistics for the PMN-4 mine at 20 cm from the antenna and for noise level of: (**a**) 35 dB, (**b**) 30 dB, (**c**) 25 dB, (**d**) 20 dB, (**e**) 10 dB, (**f**) 5 dB.

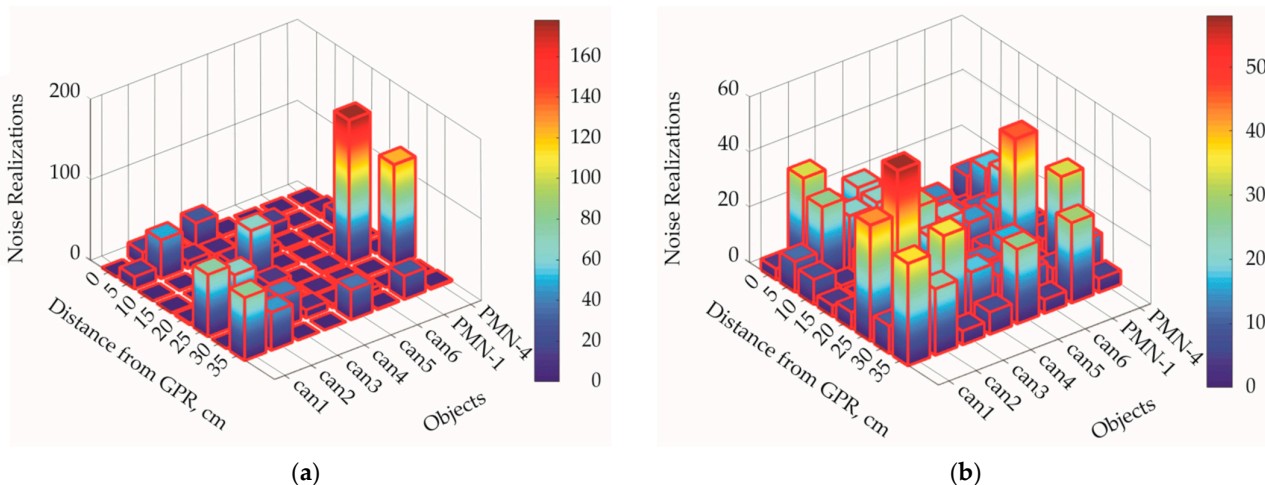

**Figure 12.** Neural network response statistics for the PMN-1 mine at 20 cm from the antenna and for a noise level of: (**a**) 20 dB, (**b**) 10 dB.

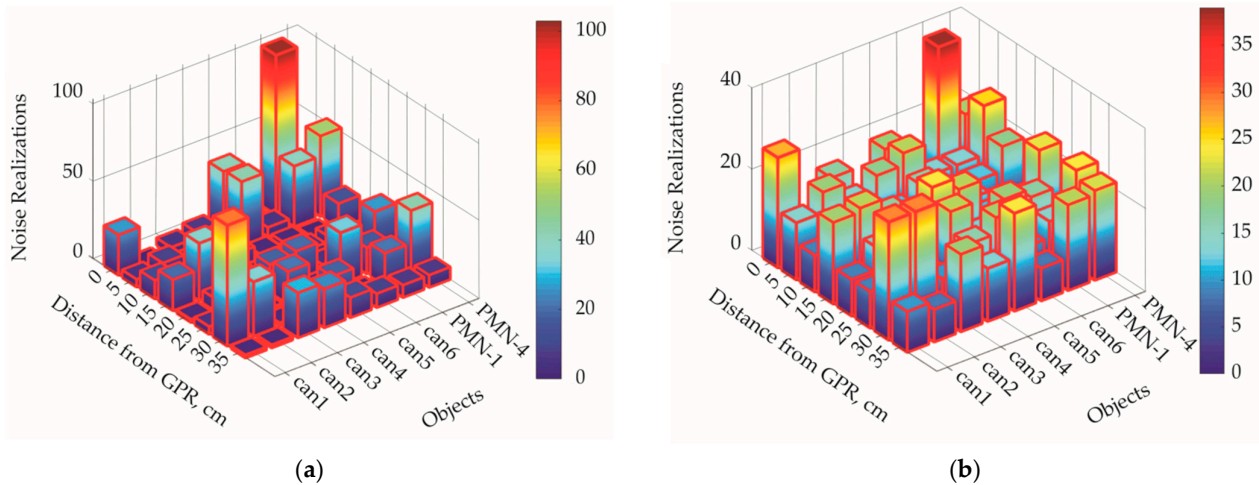

**Figure 13.** Neural network response statistics for the PMN-1 mine at 0 cm horizontal from the antenna and for a noise level of: (**a**) 20 dB, (**b**) 10 dB.

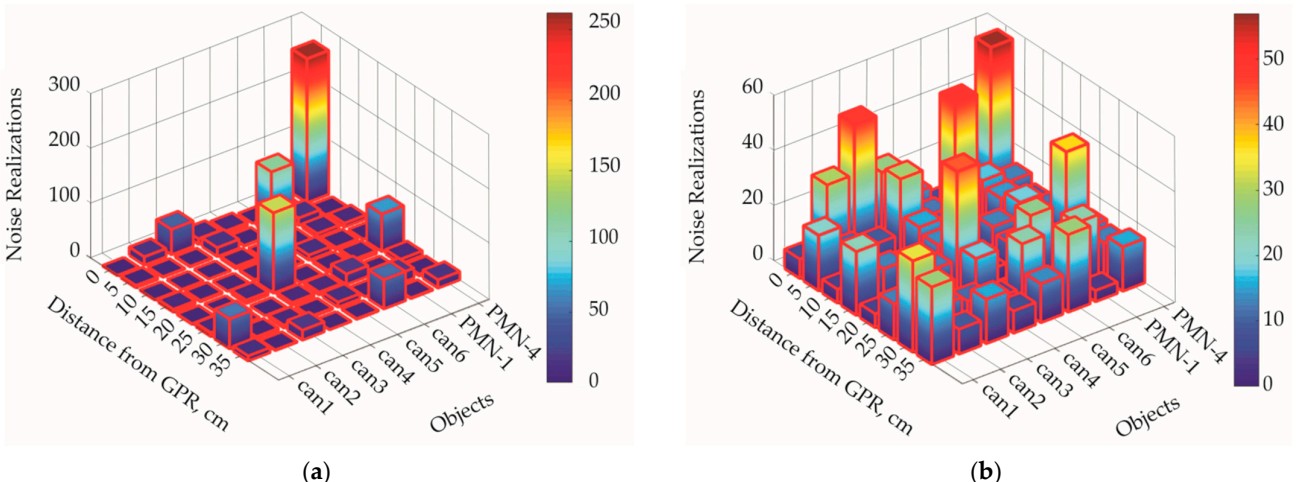

**Figure 14.** Neural network response statistics for the PMN-4 mine at 0 cm from the antenna and for the noise of: (**a**) 30 dB, (**b**) 20 dB.

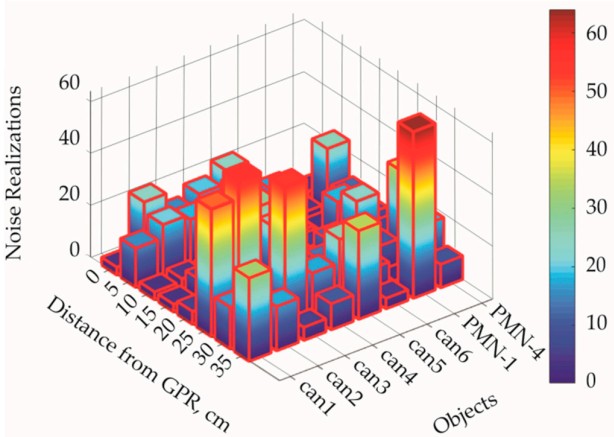

**Figure 15.** Neural network response statistics for a PMN-1 mine at a distance of 35 cm from the antenna and for a noise level of 20 dB.

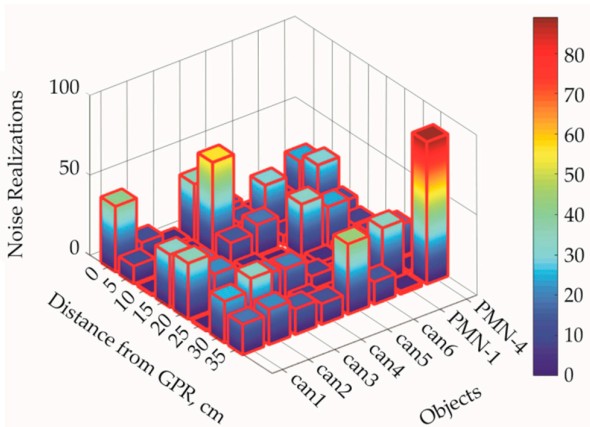

**Figure 16.** Neural network response statistics for the PMN-4 mine at a distance of 35 cm from the antenna and for a noise level of 20 dB.

### 3.2. Mine Detection in Experimental Data

The next stage of this study was applying this approach to experimental data obtained from a real GPR. The 1Tx + 4Rx antenna system [40] mentioned above (Figure 2) was installed on the robotic platform, as shown in Figure 17. Principles of action of the antenna system are described in [45]. In order to increase the radiated signal power, the overall dimensions of the transmitting antenna were increased 1.5 times with respect to the transmitting antenna described in the paper.

The frequency band of the current GPR uses a frequency range 0.8 GHz and 1.6 GHz antennas of central frequency 1.2 GHz. Height of the antenna system above the ground surface at the radar measurements is 32 cm. The objects were buried in the ground. Thickness of the soil over the objects was about 3 cm. The sampling time is 10 ps. The whole amount of samples in the A-scan is 512. It means the whole time window is 5.11 ns. The length of the test path (Figure 17) is 2.6 m. The whole path consists of 513 A-scans spaced approximately 0.50 cm along the path. The distance between the objects was 60 cm.

The time dependence of the excitation voltage is similar to the one used in the numerical simulations. The pre-processing of the input signals described in paragraph 2.1 was applied to form the training data set. Metal can and simulants of both PMN-4 and PMN-1 mines were used as objects of interest in the experimental investigation. The objects were buried in the ground at a depth of 3 cm in a linear path, with a separation of 60 cm between objects, as shown in Figure 17.

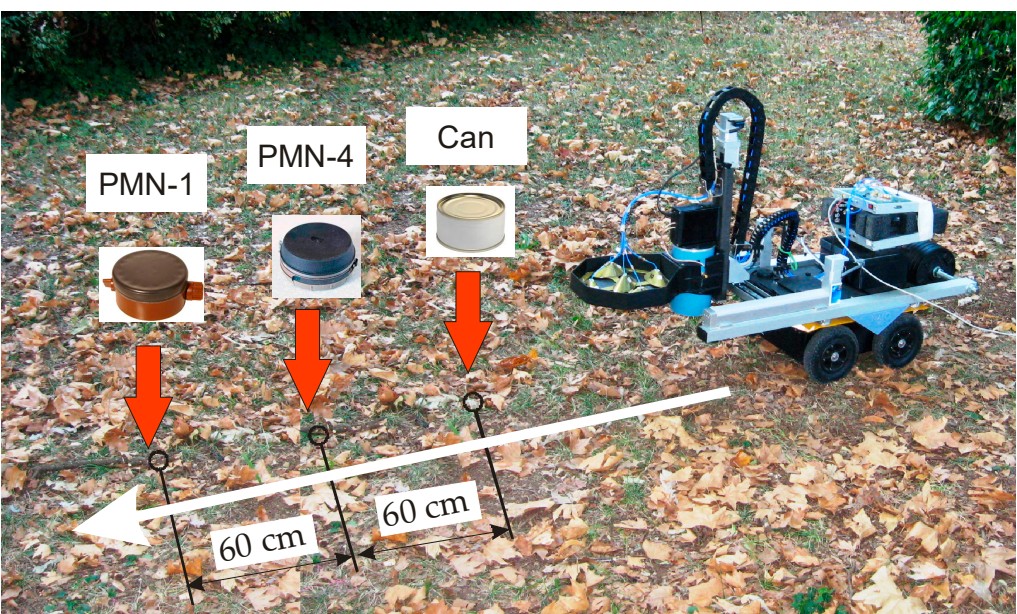

**Figure 17.** The robotic platform on the linear test path.

A fully connected neural network (Appendix B, Figure A3) with the structure 1052-500-250-94-SoftMax with neurons with a linear activation function was used to work with the experimental data. Here the 1052 input neurons correspond to the number of discrete values of amplitudes of sequentially cross-linked signals from four receiving antennas taken with a constant time step. Finally, 94 classes of mines and their positions of ANN were output, where the first class corresponds to the position of a metal can directly under the antenna, and the 2nd–31st classes point to distances of 0–18 cm for the can. The same enumeration is used to determine the initial classes for the PMN-4 target as classes 32–62 and for PMN-1 as classes 63–93. Class 94 points to the absence of any object. The SoftMax layer application allows conversion of the numerical values of the source neurons to the probability of an object being present. All input data were linked to the corresponding distances for the convenience of analyzing the received results. To verify the operation of the ANN, input data were collected by passing a robotic platform over the objects. The distance covered in the experiment was 2.6 m.

The input data were modified with Gaussian noise with different SNR ranging from 35 to 10 dB (see Figures 18–23). The distance from the object to the geometrical center of the antenna system is shown in the figures. The SNR was calculated according to the raw data, and the addition of noise allows us to assess the stability of the recognition. As a result, each graph shows the average results for 1000 implementations of noise generation. The case of SNR = 35 dB is shown in Figure 18. This level of interference corresponds to the normal mode with little distortion of the input signal. Therefore, all objects were determined unambiguously. We note the convenience of having the 94th class, which points to the absence of any object. All values in the graph are normalized to be between 0 to 1, so they can be interpreted as the probability of an object being present at a certain distance. These illustrations show that successful recognition of the object is unlikely or impossible at SNR = 10 dB and lower.

Figures 18–23 present the results of ANN testing on experimental data. The network has 94 output neurons. Signals from these neurons refer to one or another object. For example, the 94th output in the figures is marked as "Absence". When it gives a value of "1", it means that there is no object of interest in the field of view of our antenna system. Similarly, outputs 1–31, 32–62, and 63–93 inform about the detection of a specific object at a specific distance from the geometric center of our antenna system. All values in the graphs have a value from 0 to 1, which can be interpreted as the probability of detection of this or that object.

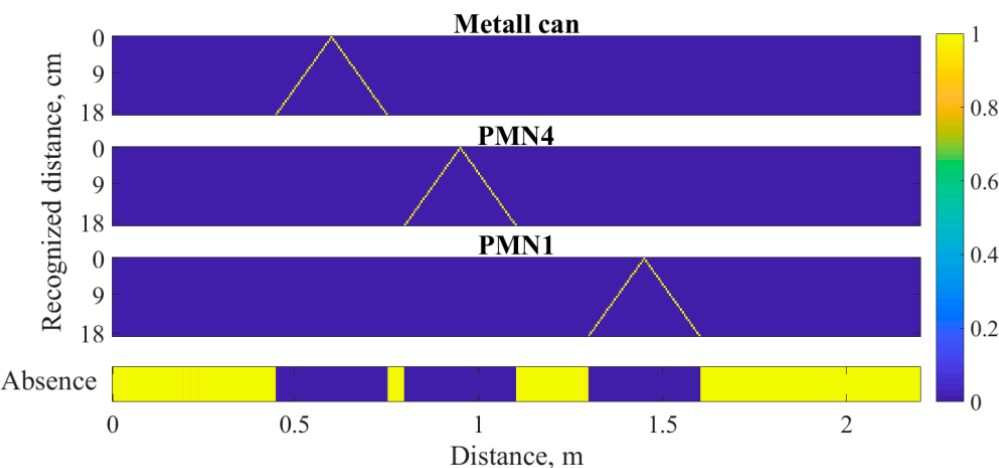

**Figure 18.** The average values of the recognized objects and their corresponding distances with SNR = 35 dB.

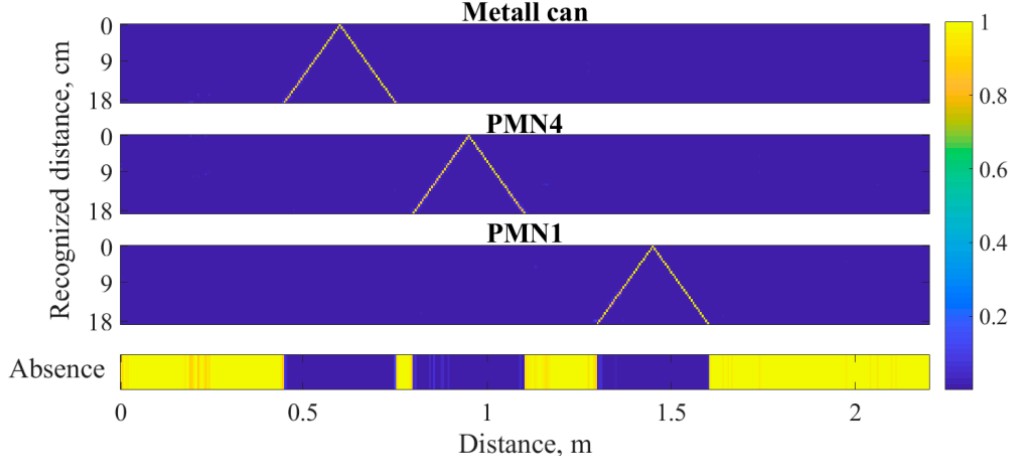

**Figure 19.** The average values of the recognized objects and their corresponding distances with SNR = 30 dB.

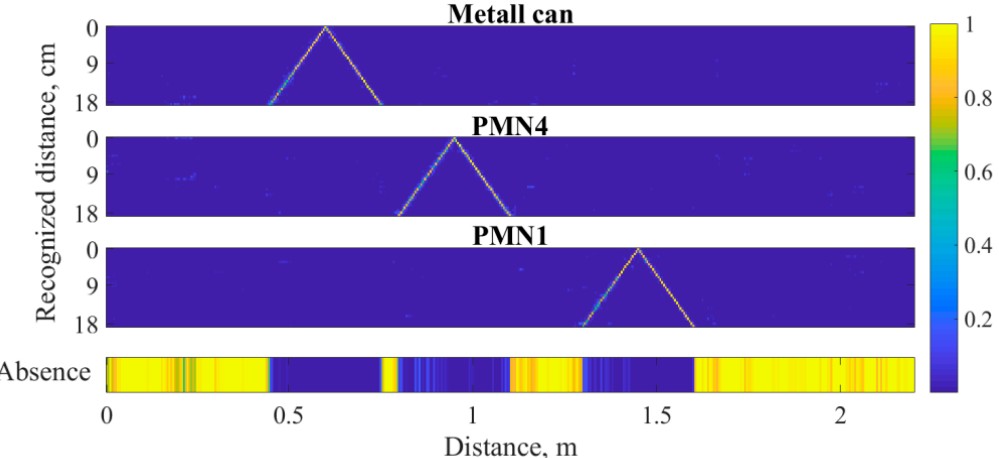

**Figure 20.** The average values of the recognized objects and their corresponding distances with SNR = 25 dB.

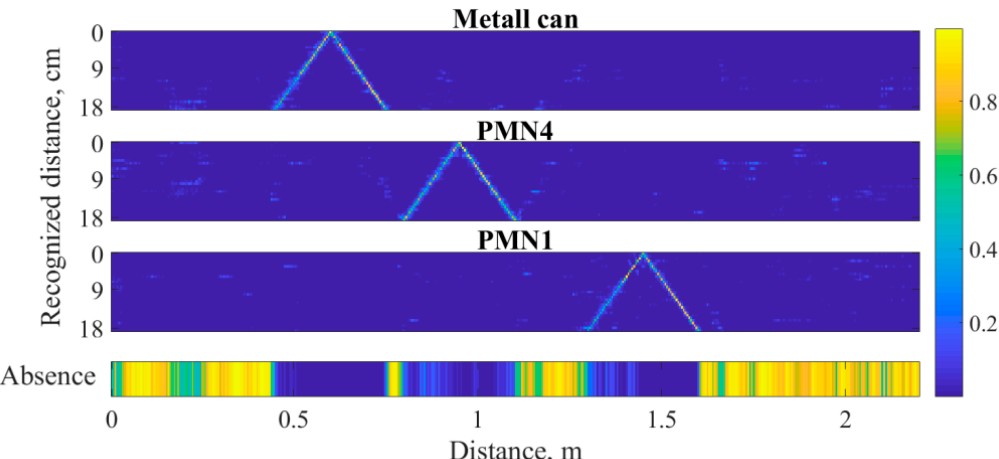

**Figure 21.** The average values of the recognized objects and their corresponding distances with SNR = 20 dB.

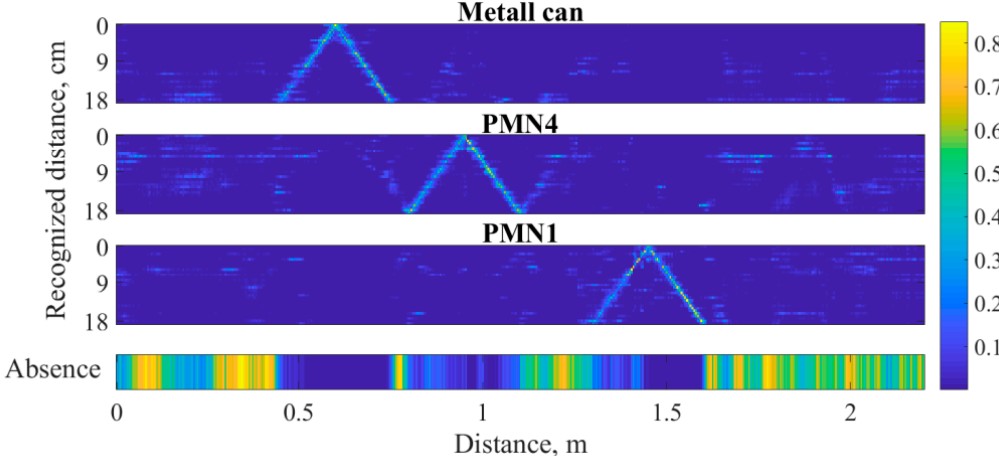

**Figure 22.** The average values of the recognized objects and their corresponding distances with SNR = 15 dB.

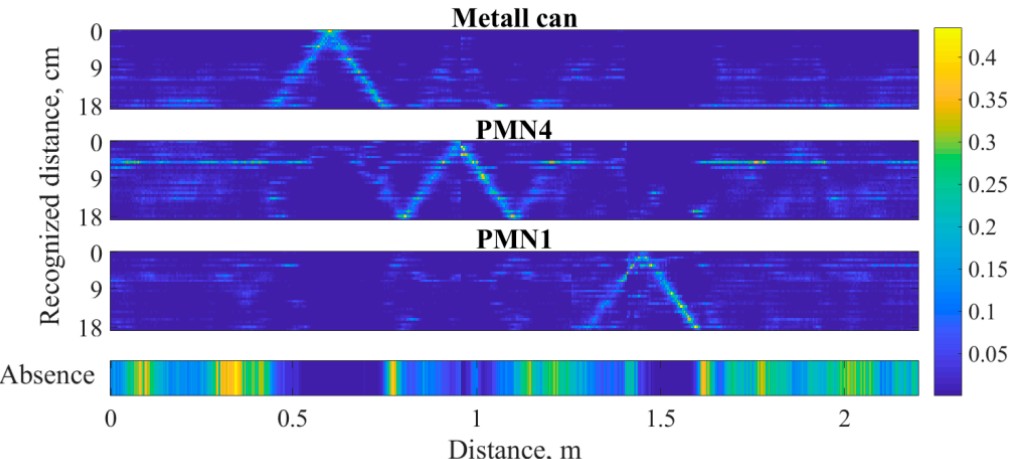

**Figure 23.** The average values of the recognized objects and their corresponding distances with SNR = 10 dB.

### 3.3. The Problem of Recognizing Dielectric Objects in the Sector of Angles

In our previous simulation, we considered the case where the hidden objects were located only along a straight line corresponding to the direction of the GPR movement.

However, this is an ideal case which is not common in reality; an object can be located at different angles relative to the robotic platform. Therefore, we improved the simulation by adding another dimension in which we place objects at various angles relative to GPR, as the case is with actual subsurface surveys. We consider a schematic model of the investigated area for the given problem. A polar coordinate system is superimposed on the investigation area, the center of which is where the irradiating antenna is (Figure 24). Since we are only looking for objects in the space in front of the antenna, we limited ourselves to a sector between 30 to 150 degrees (as defined in Figure 24) with a step size of 20 degrees. The horizontal radial distance from the antenna to the object ($\rho$) was taken from the position of the mine under the radiating antenna (the origin) out to the maximum distance of 35 cm, in steps of 5 cm. The black dots in Figure 24 show the 38 possible positions of the mine with these two coordinates. In this way, we approximated the survey area, and the object's position is visually determined. We note that it is possible to take smaller steps between the possible object locations. However, this leads to an increase in the number of simulation data samples and increases the simulation time.

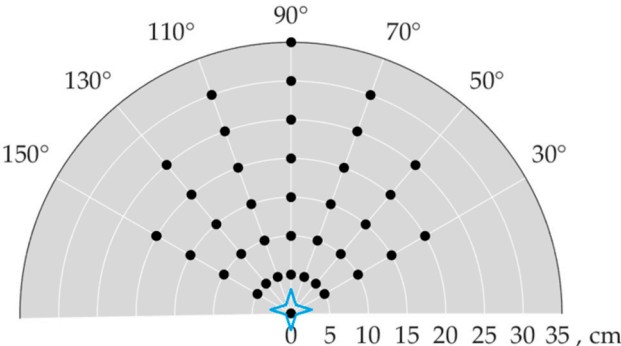

**Figure 24.** Schematic model of the area under investigation. The blue four-pointed star shows the position of the center of the 1Tx + 4Rx antenna system. Black dots mean the positions of objects in modeling the data for training ANN.

With this configuration, the problem becomes more complicated than in the previous linear situation. We therefore chose a more complicated structure for the neural network, which has a 4500-4000-2000-1000-500-2000-4000-191 layer structure in this case. This architecture has been shown to have the best recognition properties for the grid search. The input data are prepared the same way as in the previous simulation. The training sample is a sequence of 4500 time points which contains six signals of 750 time points each. It corresponds to the reflected pulse with a total duration of 7.5 ns and a time step of 0.01 ns. We obtained 191 sequences with the following considerations. There are five objects under investigation, namely a metal can, a PFM mine with longitudinal orientation, a PFM mine of with cross orientation, a PMN-1 mine, and a PMN-4 mine. We can see from Figure 24 that there are 38 possible positions for each object. So, 38 × 5 = 190, and an additional output is needed for the case where the object is absent. Each of the 191 output neurons shows the presence of an object, the object type, distance to the object, and its direction in polar coordinates. We were also motivated to use only one can instead of six. We saw from previous simulations that cans with the cut slot interfere the most with reliable mine detection. Thus, using only this model of can was advantageous for two reasons. The first was that using the most complicated case in the simulation gave us confidence that the neural network could deal with this type of can as well as the easier types. The second is that using only one can instead of six greatly decreases the number of training samples needed and saves calculation time.

Below we illustrate the effect of adding noise to the ANN input data. We can clearly see in Figure 25 the distortion of the waveforms for different signal-to-noise ratios (SNR). In each case we have six stitched pre-processed signals. Purple lines show the data with

no noise applied while the black curves show the same signal but with added noise with different SNR values.

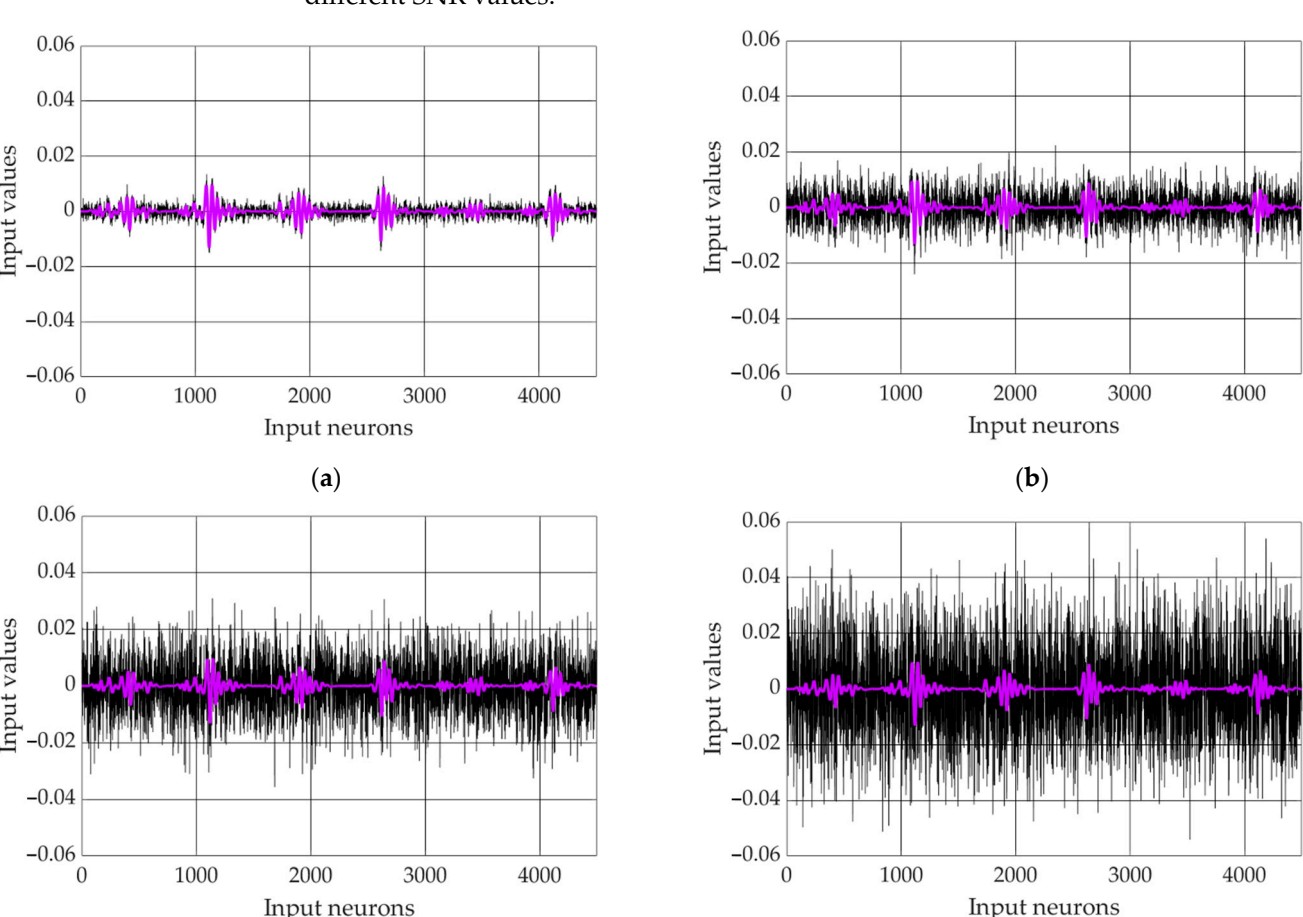

**Figure 25.** Illustration of the distortion of the data samples by applying white Gaussian noise with SNR of (**a**) 30 dB, (**b**) 20 dB, (**c**) 15 dB, (**d**) 10 dB. Purple lines correspond to the data samples without the noise. Black lines designate the data samples with additive noise.

The efficiency of the object detection by the trained neural network from the noisy input data of different SNRs is outlined below. The recognition results are presented in the form of the model of the studied space, which is defined in Figure 24. The detection results of PMN-4 and PFM from noisy input data are represented in Figures 26 and 27, respectively, where colors denote the type of object, and the size of the circle symbolizes the probability of the object being present.

For each recognition problem there were 1000 noise realizations. The statistical distribution of the ANN results is shown in Figures 26 and 27. The radius of the circle shows the sum of the ANN corresponding outputs for a given specific position from all 1000 recognition attempts. The legend contains statistics concerning all objects as a percentage of the 1000 answers obtained by the ANN. Further, 0% means that this type of object is absent in the study, and 100% means that this type of object definitely present. Data in the frame of Figure 26 shows that PMN-4 is definitely presented at the place and other objects (can, PFM cross, and others) are definitely absent at the place. Intermediate values mean probability of detection of corresponding object. Thus, the radius shows the calculated probability of the object being present.

This statistical result simulates the real-time recognition where the GPR irradiates the ground with impulses having a certain step in time. This approach provides an opportunity

to assess the resilience of the system to artificial interference. Thus, our approach should have practical value.

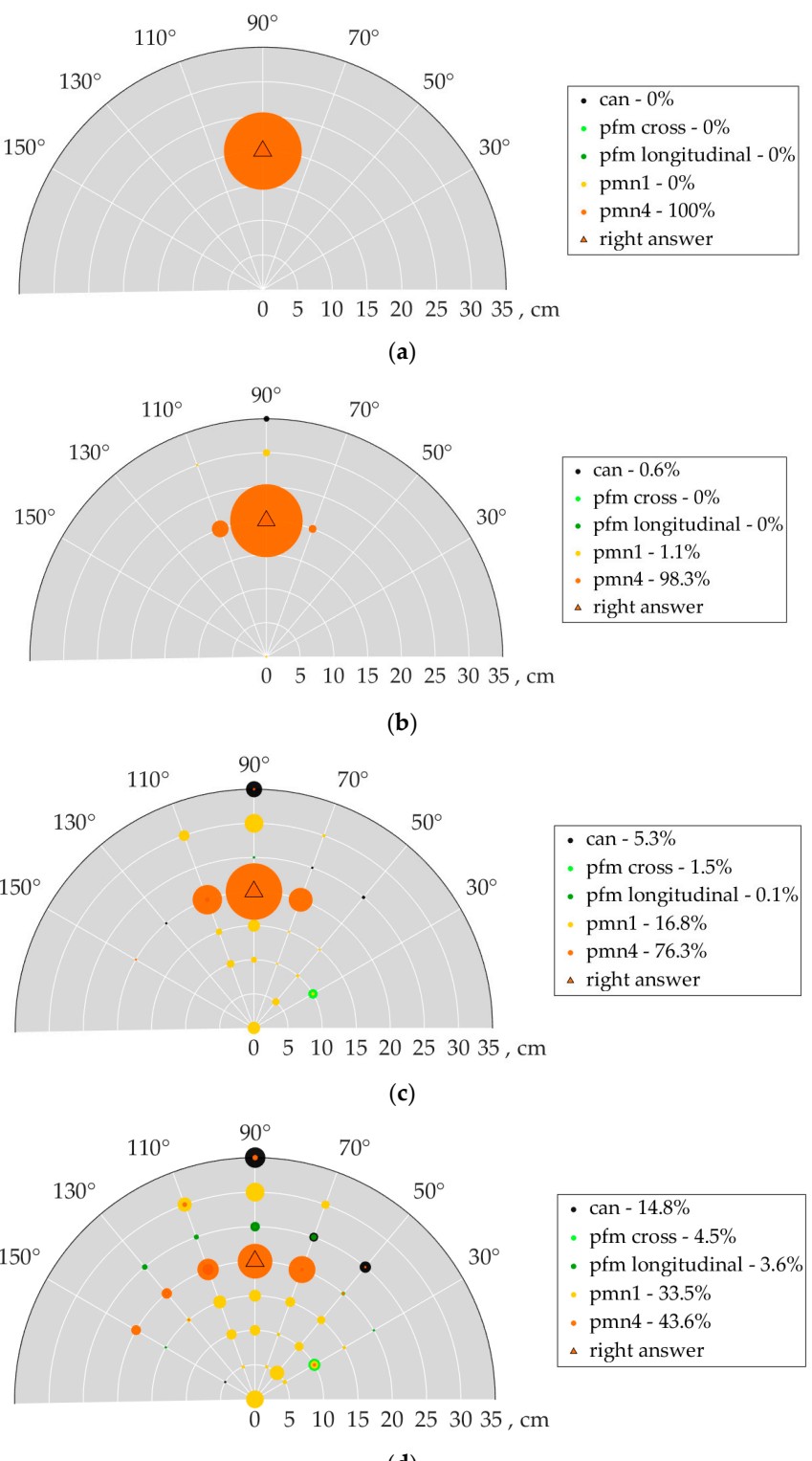

**Figure 26.** Results of the recognition of the PMN-4 mine for a distance 20 cm and angle 90° with SNR levels of (**a**) 30 dB, (**b**) 20 dB, (**c**) 15 dB, (**d**) 10 dB.

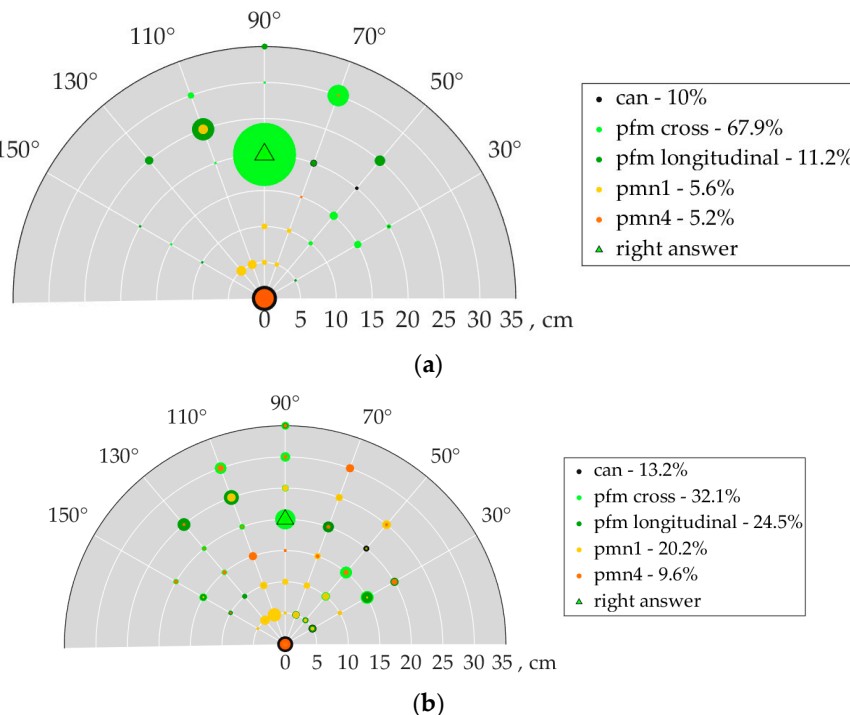

**Figure 27.** Results of the recognition of the PFM mine with cross orientation located at a distance 20 cm and angle 90° with the following SNR: (**a**) 30 dB, (**b**) 20 dB.

We see from Figure 26 that the recognition of a PMN-4 mine becomes worse at a noise level of SNR = 10 dB. However, we obtain a good approximation of the responses of the neural network around the correct position, i.e., ρ = 20 cm, phi = 70, 90, 110 degrees. In addition, the neural network is still 45% confident that the PMN-4 is located in front of the radar. Although we see that the correct object position response is beginning to deviate, incorrect answers begin to prevail. For the PFM mine, the results are worse. Determination of the correct position of this mine becomes almost impossible at an SNR of 20 dB, although at the same time the neural network is still 58% confident that it is some spatial form of the PFM mine. A weaker impulse reflection from a dielectric mine such as PFM complicates its recognition, but it is still possible for lower SNRs.

We decided not to merge the PFM spatial positions into one class, which would be a logical approach at first glance. However, this would cause a class imbalance in the training dataset. To avoid this, we have expanded the training set and divided the provisions of the PFM into two different classes.

We next consider the neural network approximation in more detail and apply it to the test data for which it was not trained. These testing samples were obtained as follows: a training position for the mines was fixed at ρ = 20 cm and φ = 90 degrees. From here it was adjusted in space in several possible ways, for example with an offset to the right or left, or with self-rotation at some angle. Thus, a variety of new positions were selected for which the ANN had not been trained before. The simulation was then performed for the PMN-1, the PMN-4, and the cross orientation of the PFM. These training cases represent the range of real conditions of an ordinary subsurface survey, where the mine is not necessarily located at our discretized nodes in the investigated space.

The neural network was only partially successful in recognizing buried mines that were rotated at an angle relative to their training position. However, a mine displaced to the side was recognized perfectly, as we show below.

The circles in Figure 28 illustrate the recognition of the PMN-4 mine, which has been translated 3 cm to the left (toward the angle 110 degrees) from the position ρ = 20 cm, φ = 90 degrees:

We see that for large values of the SNR, the ANN is more certain in object identification at the node at ρ = 20 cm, but it makes small errors in angles. However, with the addition of larger noise levels, we can trace interesting dynamics in the approximation of the response, i.e., the radii of the large circles begin to equalize. This indicates that the total number of responses of these most active output neurons of the neural network are also beginning to equalize. From Figure 28e we estimate a roughly equal likelihood of identification of that mine at ρ = 20 cm, φ = 90° or at ρ = 20 cm, φ = 110°.

In Figure 29, P is the point that represents the ρ = 20 cm, φ = 90° node in our space discretization, P1 is the point at ρ = 20 cm, φ = 110°, O is the point under the irradiating system, namely ρ = 0 cm, and P_test is the actual mine location. Thus, in this test case the PMN-4 mine (P_test) is at a distance of X = 3 cm from the position of ρ = 20 cm, φ = 90° (point P) and is at a distance of z = 4.025 cm from ρ = 20 cm, φ = 110° (point P1), i.e., it is almost equidistant. Therefore, the evenly split response of the ANN in the two positions in Figure 29 at a large SNR value can be interpreted as the position in the middle between the two most active neurons. It is an ideal ANN response which corresponds to the intermediate position in this test case.

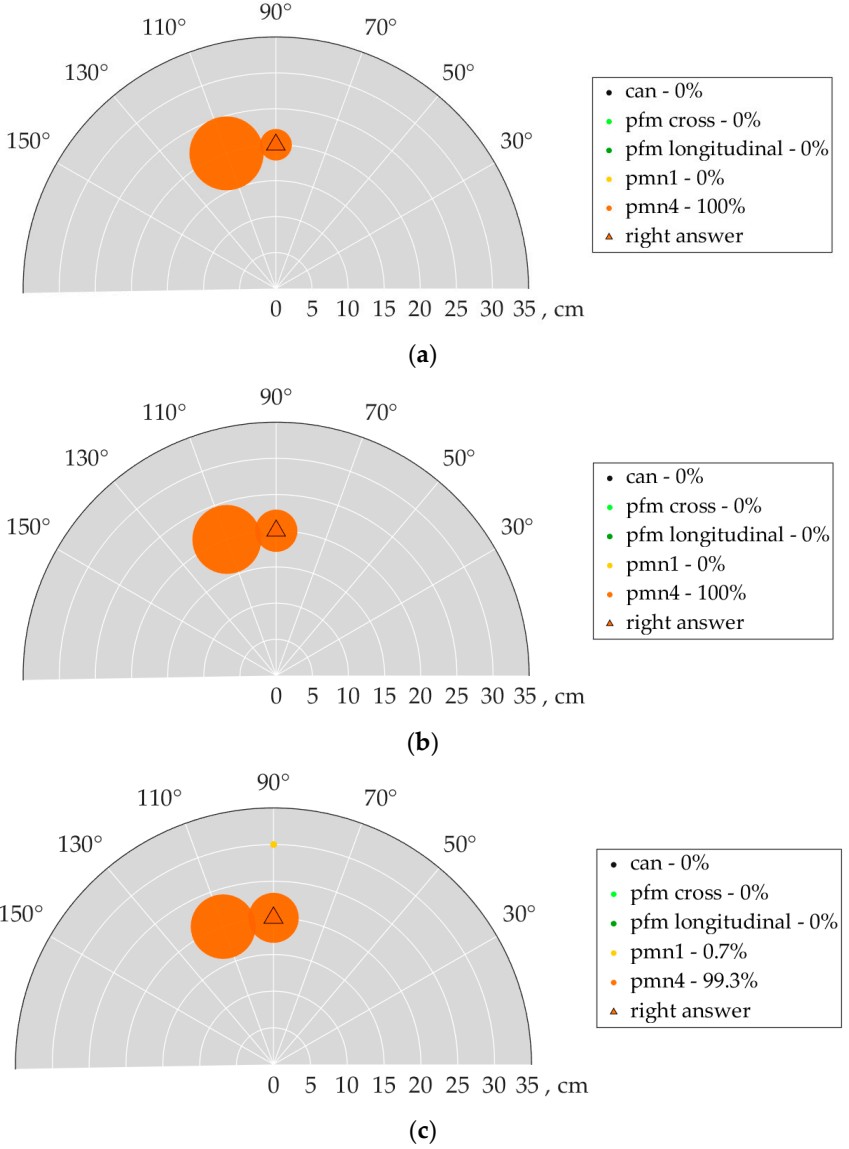

**Figure 28.** *Cont.*

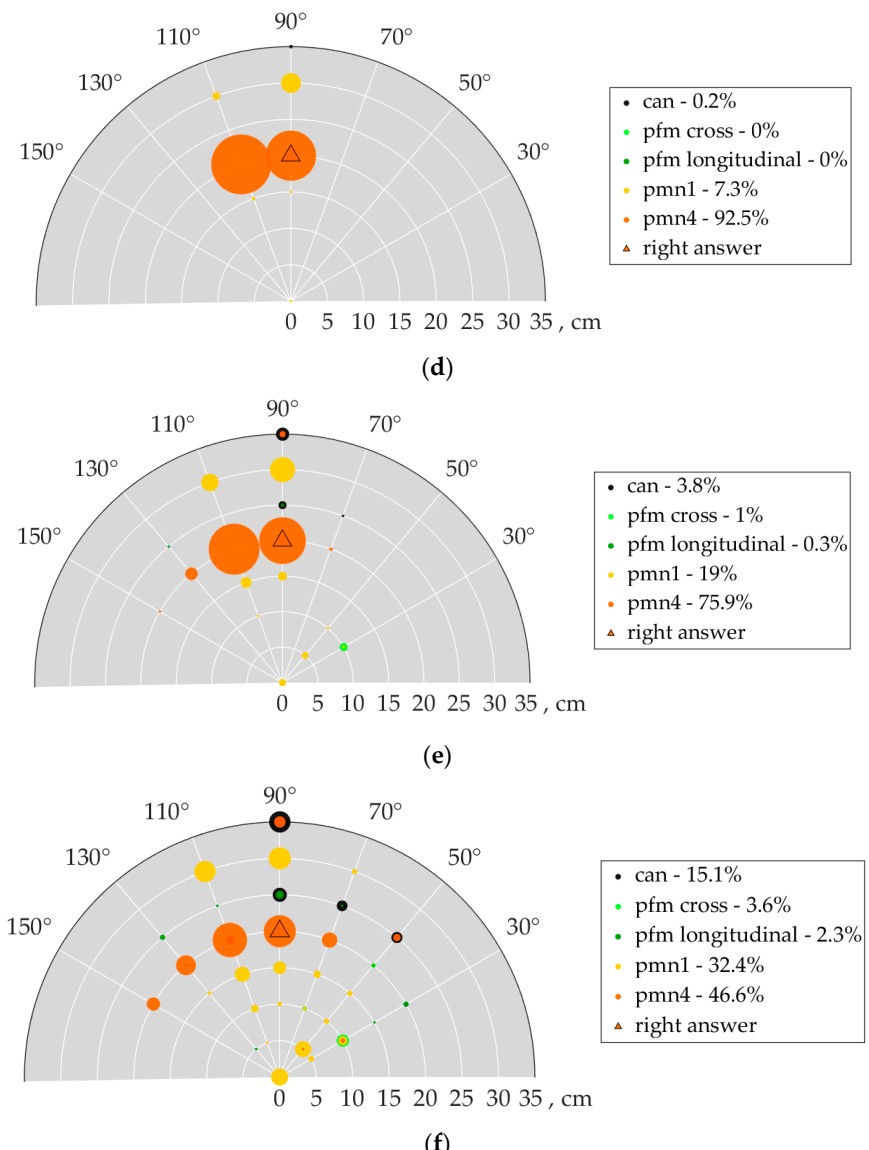

**Figure 28.** Results of the recognition of a PMN-4 mine, which is deflected guidance, will be provided for full proposals, which will be due 3 cm left from the training position of ρ = 20 cm and angle 90° with the following SNR: (**a**) 35 dB, (**b**) 30 dB, (**c**) 25 dB, (**d**) 20 dB, (**e**) 15 dB, (**f**) 10 dB.

We also note that the spatial deviation of 3 cm in the test data was a maximum. Smaller values of the deviation of the mine from the starting position at ρ = 20 cm, φ = 90° are classified by the neural network as ρ = 20 cm, φ = 90° without adding noise. However, the addition of noise shifts the points of recognition.

### 3.4. Improvement of the Data Processing Algorithm with a Neural Network Ensemble

As described above, we improved our investigation of the target area using approximately real conditions of a subsurface survey, but we need to improve our machine learning approach as well. In our three previous simulations we used a fully connected neural network. This is a good choice for simple tasks, but with our more complex simulation conditions and more complex testing cases, our machine learning approach also needs to be improved. There are a number of modern approaches for artificial intelligence problems. For example, we can compare our results with the ensemble learning technique.

We first define the input and output layers for all networks from the ensemble. For an input we can still use our previous pre-processing algorithm since it was sufficiently

successful in our previous simulations. For the output of the neural network, we have 13 neurons, 12 of which signal information about the object at a certain distance. We have three objects: the metal can (with cut slot), the PMN-1, and the PMN-4 at four training distances: 0, 10, 20, and 30 cm. There is also a neuron which indicates the absence of an object in the investigated area.

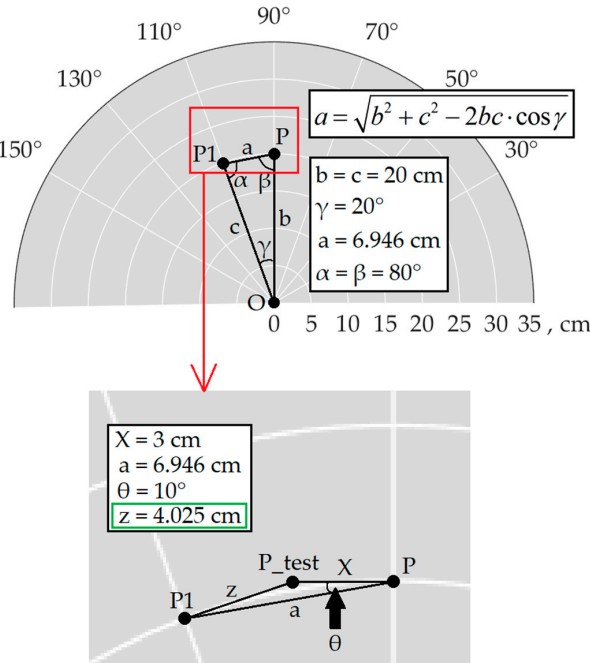

**Figure 29.** Geometric determination of the real spatial position of the PMN-4 mine. Magnified part of the polar plot with dots P1 and P denoted by red frame is shown below.

We now consider the structure of the ANN ensemble (Appendix B) for this simulation. All the neural networks can be divided into two levels, as shown in Figure 30. Each network in the first level takes the data sample and outputs its result independently of the others. The "Supreme" network of the second level receives the answers of the previous networks, and it forms the final answer of the whole ensemble. This is a classical stacking method.

In the first level we see a fully connected neural network (Figure A3). It has the structure 4614-1024-512-256-13 and the ReLu activation function (Figure A4). A Recurrent Neural Network (RNN), Gated Recurrent Units (GRU) and Long Short-Term Memory (LSTM) networks were also used. Each of these has a layer structure of 4614-512-512-13 and uses the hyperbolic tangent as an activation function. The supreme network is fully connected and also has a simple structure, namely 4-512-256-13. The supreme network has a ReLu activation function as well. The input size is four because the transformation from category to units is made. That is, each categorical response of the 13 values of the output neurons of the first-level networks is transformed into a specific class called a unit. Since we have four networks, we have four answers that are input into the second-level network. The output of the supreme network gives the final answer.

The training of this ANN ensemble has interesting features. Since one of the purposes of this work was to check the noise immunity of the machine learning approach, for this ensemble we artificially altered the data by adding noise with certain levels to the calculated ideal time dependences. This is classic data augmentation and solves two problems at once. The first is the inability to control ANN learning and validate results, and the second is the lack of training data. Due to the random nature of noise, we acquire unique examples for use in the learning procedure. Additionally, this approach is expected to increase the noise immunity of the ensemble, since a sufficient number of training samples were generated and good validation accuracy (about 90%) was obtained. It demonstrates the ability of the

ANN to generalize the noisy data. As a result, there are 1000 examples with a noise level distribution of 35–15 dB for each class.

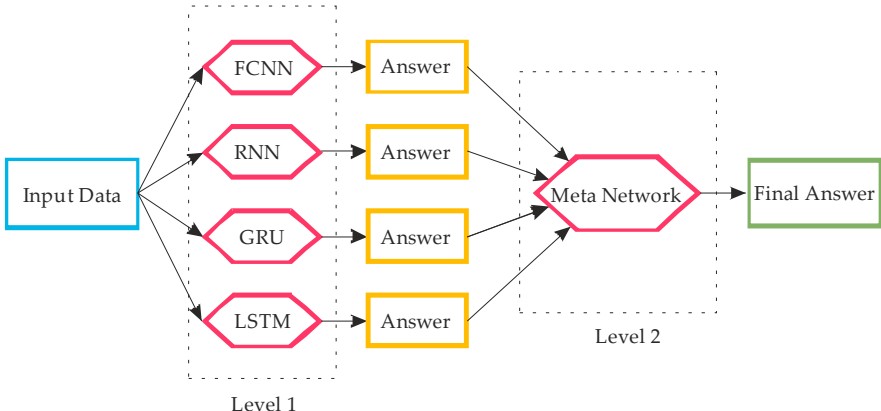

**Figure 30.** Structure of our stacking ensemble.

The training of the supreme network was realized separately. An input dataset for it consisted of the recorded responses from the first-level networks on data having different noise levels (from 35 to 15 dB). After receiving good validation results, the networks could be connected as shown in Figure 30, so that the ensemble was ready for testing.

For a qualitative check on the functioning of the ensemble, it is necessary not only to see the final answer but also the answers from the first-level networks. The main purpose of ensemble building is to illustrate the advantages of the collective approach in comparison with the single answers from the first-level networks (Appendix C).

We calculated 500 or 1000 realizations of recognition for each test case to obtain a reliable statistical answer in our previous simulations. However, in this analysis we decided to increase the number of implementations to 10,000, since it gives even more reliable and statistically precise results. One can estimate the relative quality of recognition and a statistical scatter of the answers for each of the networks in Figures A5–A10 with the help of a color bar. For accurate estimations, the number of hits for each neuron in each network is displayed in each cell.

Figures A5 and A6 show a comparison of the results of PMN-1 and PMN-4 mine detection at a distance of 30 cm from the GPR system. First, we see that data augmentation for this problem is justified, since Figure A5 shows a fairly clear answer for PMN-1 up to SNR = −5 dB, while for the PMN-4 mine in Figure A6 the limit is SNR = 5 dB. This is a rather successful improvement in the noise immunity, which we did not achieve in our previous simulations. It should also be noted that this tendency is observed for all recognition results for any object (Appendix C). In viewing the main tendency of the responses of all neural networks in Figure A5, there is an evident predominance of supreme network performance. We see that even at the extremely high noise level of −5 dB, it has the biggest number of hits in the correct neuron.

In addition, one can consider the deviation of the answers for all networks, especially the neurons that correspond to the metal cans at all distances. It is clear that the supreme network is the least prone to the most undesirable responses, such as the metal can, and it is less accurate in determining distances and identifying other mines, which is not as critical. This result can certainly be considered successful since it justifies the use of the ensemble.

However, a less successful result is shown in Figure A6 for the PMN-4 mine recognition. Here the performance of the supreme network equals the performance of FCNN and RNN. Additionally, the noise immunity was slightly weaker than in the results presented in Figure A5. However, in general, the spread of incorrect answers from the supreme network is still more focused on the less-undesirable cases compared to the answer spread seen

for the first-level networks. This confirms the better performance of the neural network ensemble compared to a single network.

The results from the recognition of PMN-1 and PMN-4 mines at a distance of 10 cm are presented in Figures A7 and A8. One notices that in some cases the supreme network loses in performance and accuracy compared to the networks in the first level. However, there is still obvious resistance to the more critical cases of can recognition instead of mines. This is especially evident in Figures A7c and A8c.

Previously, we did not investigate the performance of the ANN with an empty area, and it is important to determine a noise threshold for which detection can be considered reliable. However, it is first necessary to investigate the reaction of the ensemble to the presence of the metal can in the investigated area and, secondly, to investigate the results for the case of no object present.

First, we consider the recognition of the metal can at a 20 cm distance from the GPR, as presented in Figure A9. The metal can has the strongest reflection of the electromagnetic field, and it is more easily detectable in the received signals than the other investigated objects. However, the noise immunity in this case is similar to that of mine detection (Figures A5–A8). This implies that the neural network processes these time dependences not just on the signal level, but also on its unique features. Weak reflection from an object with a higher content of dielectric components usually complicates recognition due to the weaker reflection, but the structure of our neural network ensemble is chosen so that this complexity is addressed.

Next, we consider the case of an empty investigated volume with no objects present. The recognition results are shown in Figure A10. For a noise level of 20 dB, we observe a tendency toward identifying the correct neuron. However, at the level of 10 dB, the ANN produces only a uniform distribution of responses across all neurons, and the recognition ability is lost in this case. However, this result has a very interesting application. If we look at the neural network performance for a noise level of 10 dB for the case where some object is present, then we do have a clear detection of an object being present in the investigated area. This result can be interpreted as follows. During a real subsurface survey, noise of approximately this same level is an integrated part of the received signal. If we do not obtain a clear answer for the recognition with this level of noise present, then this reliably indicates that there are no objects in front of the irradiating system, because according to the results in Figures A5–A9 we should obtain some definite indication of some object being present. Therefore, we can conclude that our approach is quite successful in detecting the presence of objects.

## 4. Conclusions

The use of an artificial neural network for GPR object recognition makes possible the detection and classification of subsurface objects. In the paper we demonstrated good noise immunity for different trial object distances from the antenna system. Using the neural network as an analyzing system leads to stable results in terms of antipersonnel mine detection, in spite of the pre-processed data being highly noised.

Probabilities of detection and false alarm depend on the threshold that we choose to declare a detection. Quantitative estimation can be performed using the figures in the paper.

For object positions that are between discretized spatial points, the neural network has demonstrated the ability to approximate the real position with multiple responses around the actual object position. This ensemble learning gave promising results with the reduction of false positives.

The meta network acted as a kind of smoother for the responses of the other networks since it was less accurate in some cases, but the spread of responses was more optimal.

Using the 1Tx + 4Rx antenna system in a UWB pulse GPR to collect data for an ANN and stitching combinations of sums and differences of signals to use as data for the ANN have significantly improved the noise immunity of the neural networks. As a result, the

noise threshold at which the detection of objects can be considered reliable is determined in our experimental analysis. This will be valuable when using our approach in actual GPR surveys. This work also provides some guidance on how to improve the machine learning approach to the problem of underground object detection.

**Author Contributions:** O.A.P.: methodology, software, writing—original draft preparation, V.P.: methodology, software, investigation, O.M.D.: conceptualization, funding acquisition, supervision, G.P.P.: validation, funding acquisition, project administration, L.C.: funding acquisition, conceptualization, V.P.R.: investigation, resources, performed the experiments, F.C.: writing, review, and editing. All authors contributed to the visualization of the data and results. All authors have read and agreed to the published version of the manuscript.

**Funding:** This research was partially funded by Project G5731 "Demining Robots" of the NATO/OTAN Science for Peace and Security (SFPS) and partially funded by Ministry of Education and Science of Ukraine, grant number 0120U102309. The authors wish to acknowledge the financial contribution for the publication of this paper provided by the Department of Information Engineering of the University of Florence.

**Data Availability Statement:** Data and code from this research will be available upon request to the authors.

**Acknowledgments:** The authors are grateful to Pierluigi Falorni from Florence University (Italy) and Timothy Bechtel from Franklin and Marshall College (USA) for fruitful discussion on the application of ANN for mine detection.

**Conflicts of Interest:** The authors declare no conflict of interest.

## Appendix A

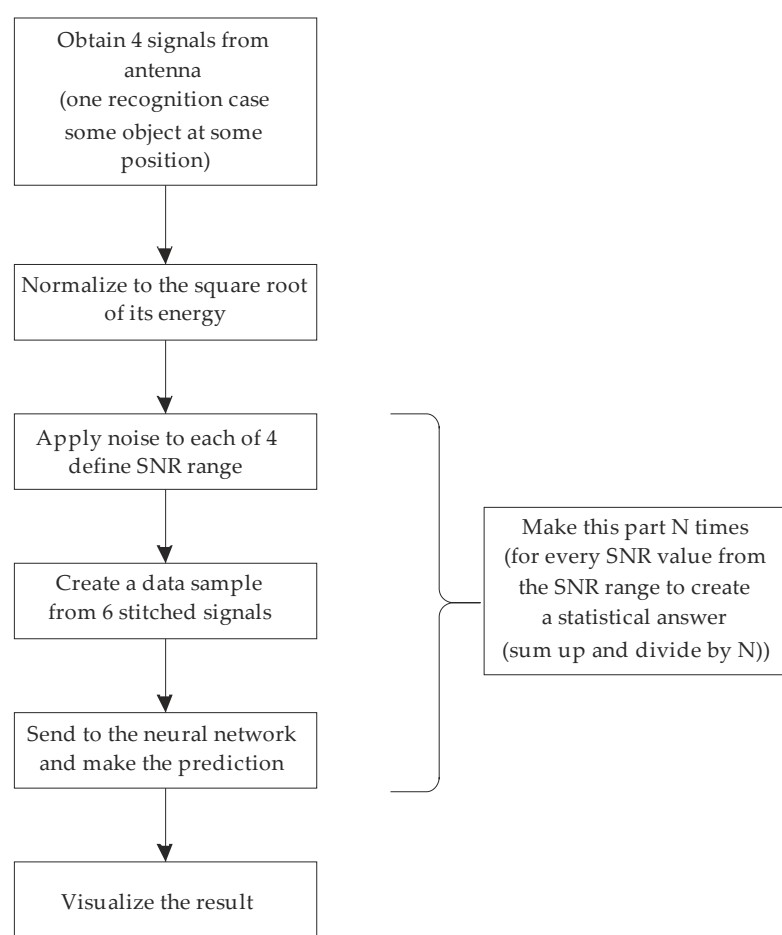

**Figure A1.** The flow chart of the work pipeline.

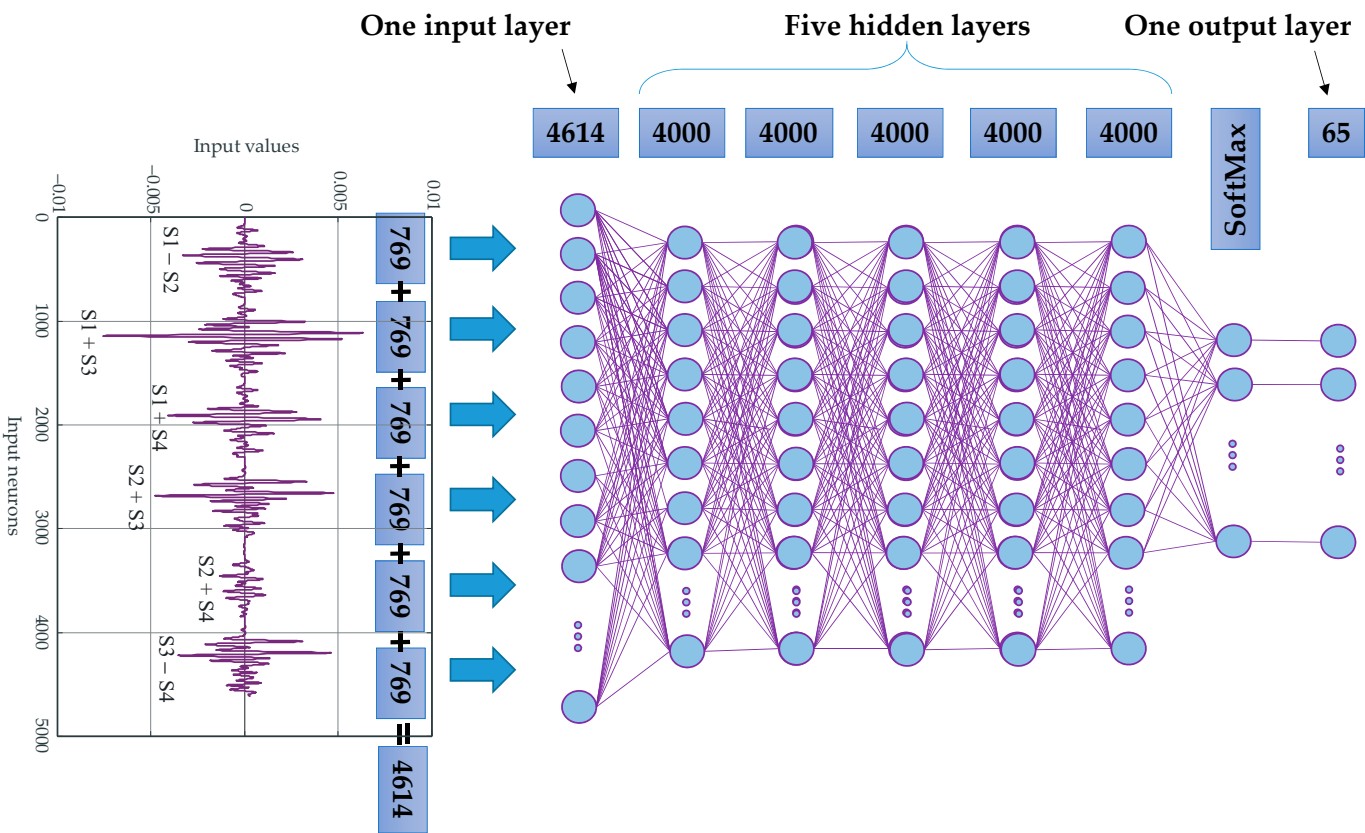

**Figure A2.** The fully connected ANN.

**Appendix B**

In Sections 3.1–3.3 we used fully connected neural network.

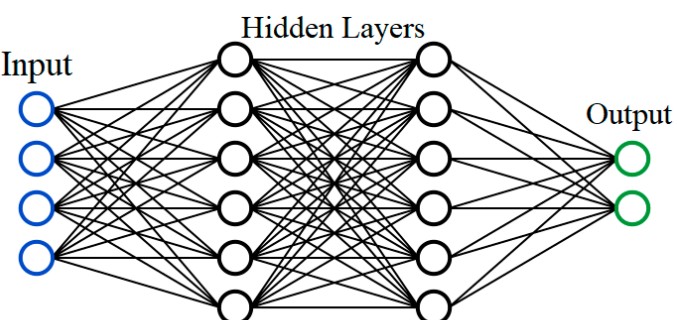

**Figure A3.** Structures of neural networks. Input layer (blue circles), hidden layers (black circles), output layer (green circles).

Section 3.1:
Input layer: 4614 neurons (4614—resulting array shape of input data sample (769 points (7.69ns) for 6 signals. $769 \times 6 = 4614$))
5 hidden layers, each of 4000 neurons
Tanh activation functions
Softmax output layer
Output neurons: 65 (we have 8 objects, 8 possible distances + object absence case. $8 \times 8 + 1 = 65$)
Section 3.2:
Input layer: 1052 neurons (1052—resulting array shape of input data sample (263 points for 4 signals. $263 \times 4 = 1052$))

2 hidden layers, 500, 250, 94 neurons
ReLu activation functions:

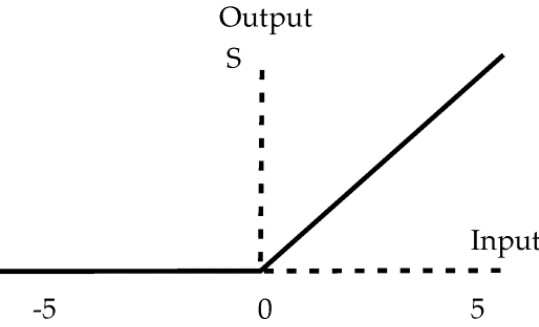

**Figure A4.** ReLu activation functions.

Softmax output layer
Output neurons: 94 (we have 3 objects, 31 possible distances for each object + object absence case. 3 × 31 + 1 = 94)
Section 3.3:
Input layer: 4500 neurons (4500—resulting array shape of input data sample (750 points (7.69ns) for 6 signals. 750 × 6 = 4500))
7 hidden layers, each of 4000 neurons
Tanh activation functions
Softmax output layer
Output neurons: 191 (we have 5 objects, 38 possible distances (angles + distances) + object absence case. 5 × 38 + 1 = 191)
Section 3.4:
Here we are using data from Section 3.1
FCNN:
Input layer: 4614 neurons (4614—resulting array shape of input data sample (769 points (7.69ns) for 6 signals. 769 × 6 = 4614))
3 hidden layers: 1024, 512, 256 neurons
Tanh activation functions
Softmax output layer
Output neurons: 13 (we have 3 objects, 4 possible distances + object absence case. 3 × 4 + 1 = 13)
RNN
Input layer: 4614 neurons
2 hidden layers: 512 each
Output neurons: 13
GRU
Input layer: 4614 neurons
2 hidden layers: 512 each
Output neurons: 13
LSTM
Input layer: 4614 neurons
2 hidden bidirectional layers,: 512 each
Dropout 0.5 for each
Output neurons: 13
META:
Input layer: 4 neurons (takes answers of previous networks in form of unit value of class number one of 13)
2 hidden layers: 512, 256 each
ReLU activations
Output neurons: 13

**Appendix C**

The data in Figures A5–A10 are shown in such a way that we can observe the whole matrix of answers at once. The OX axis depicts 4 networks of the first level and the supreme network. The OY axis shows the neurons for each network, including the supreme network. In addition, the correct answer in Figures A5–A10 is enclosed by a green frame.

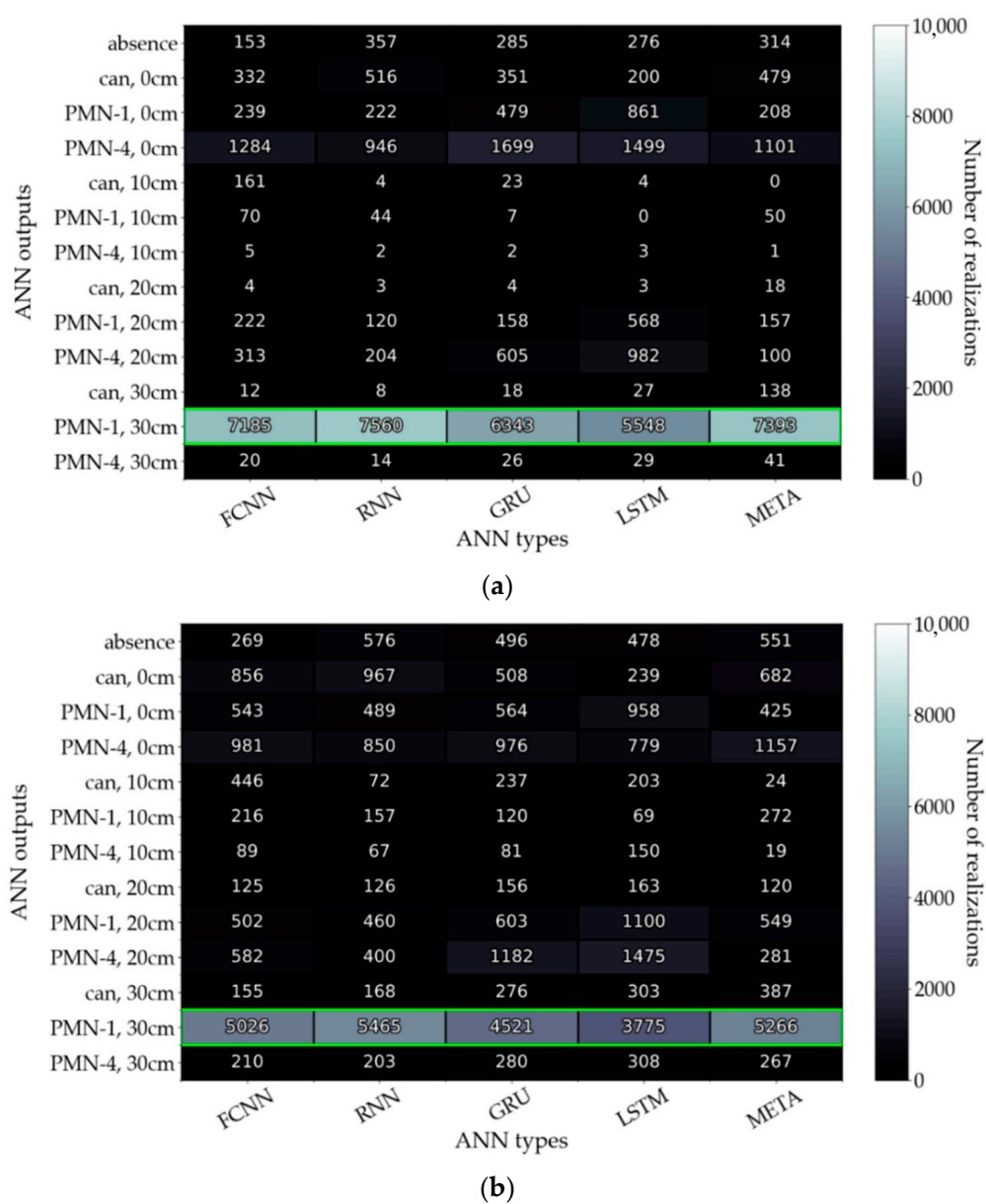

**Figure A5.** *Cont.*

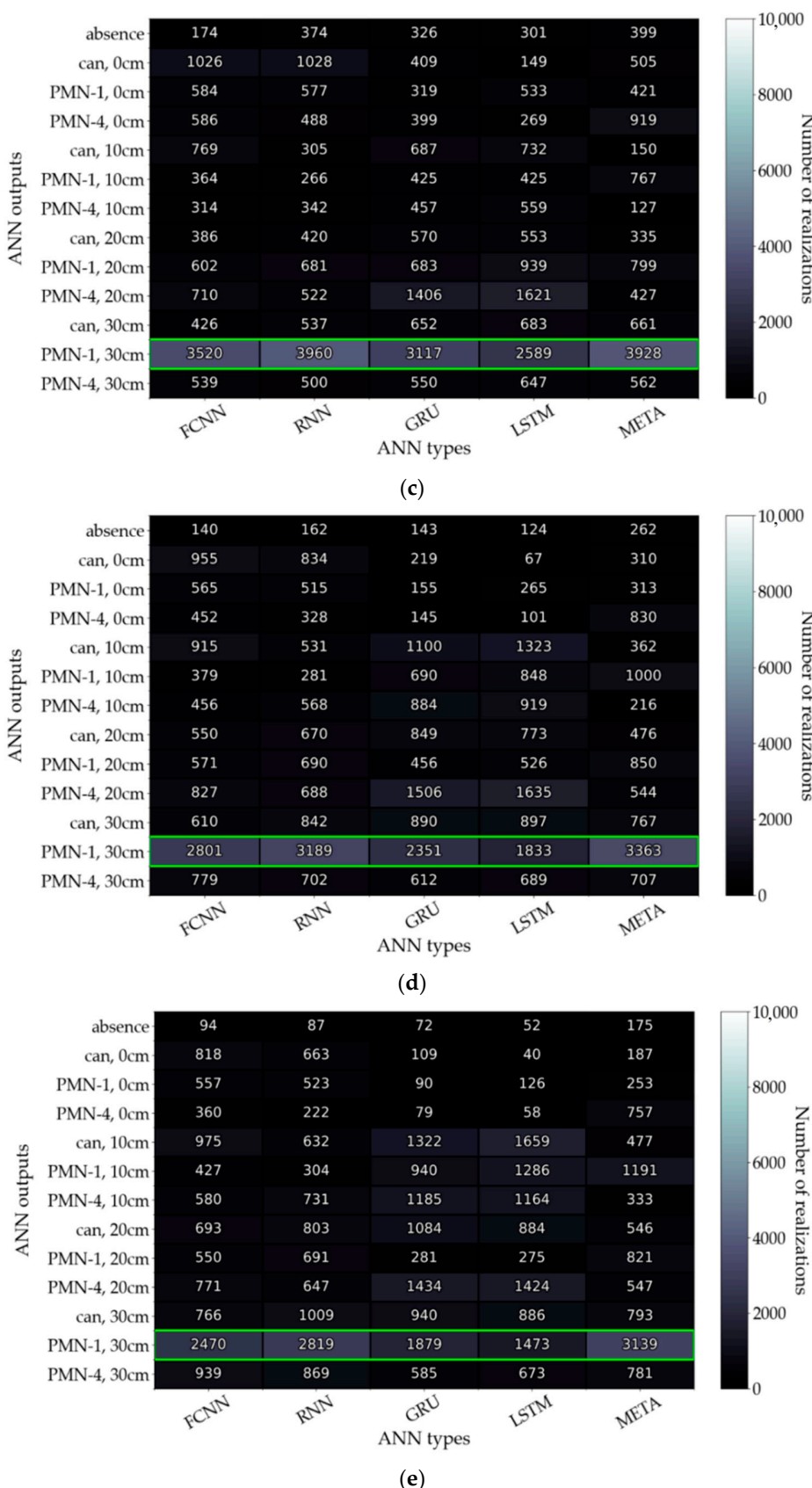

**Figure A5.** Matrix of output neuron answers of each network from the ensemble for the recognition of a PMN-1 mine at a 30 cm distance for (**a**) SNR = 15 dB, (**b**) SNR = 10 dB, (**c**) SNR = 5 dB (**d**) SNR = 0 dB (**e**) SNR = −5 dB.

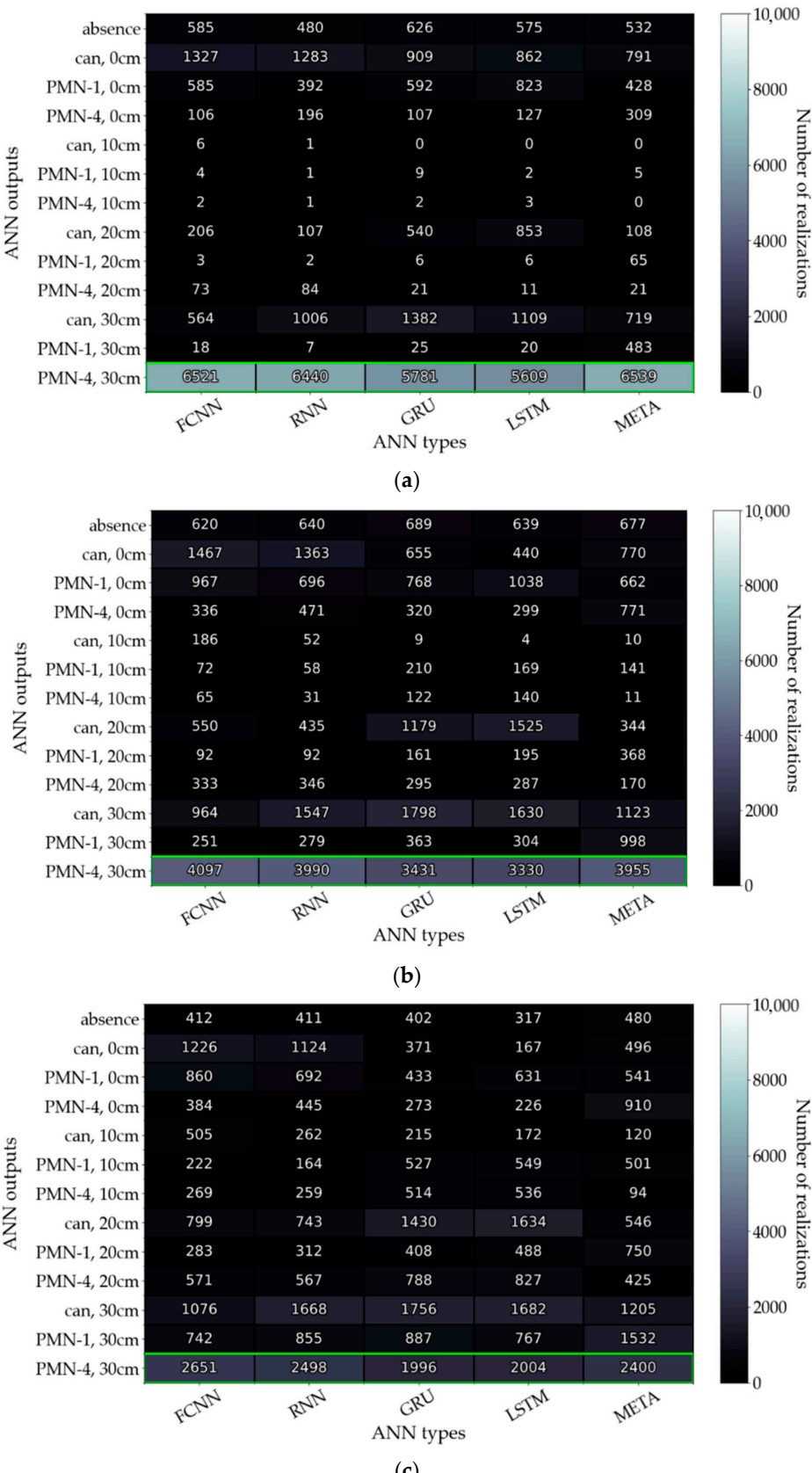

**Figure A6.** Matrix of output neuron answers of each network from the ensemble for the recognition of a PMN-4 mine at a 30 cm distance for (**a**) SNR = 15 dB, (**b**) SNR = 10 dB, (**c**) SNR = 5 dB.

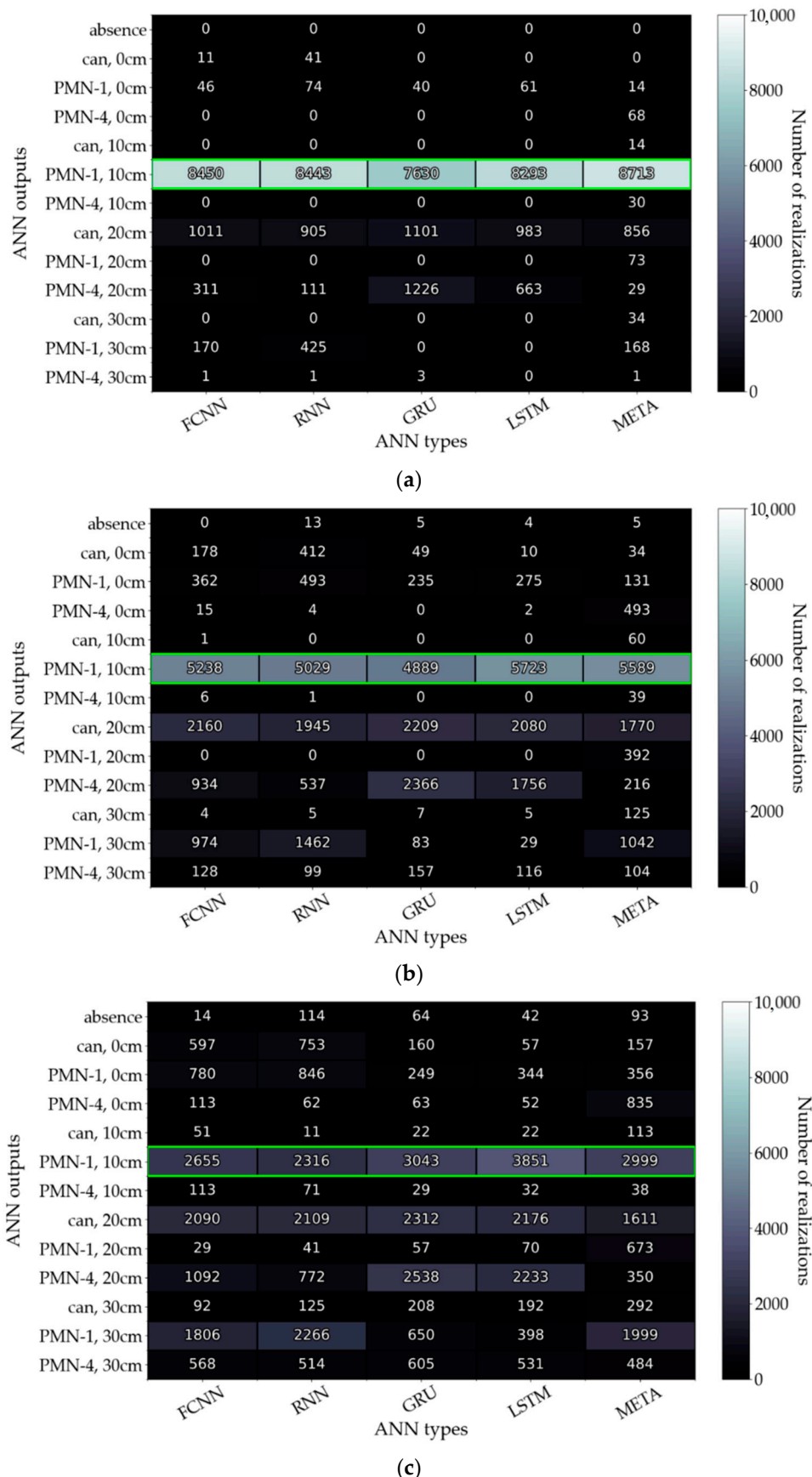

**Figure A7.** Matrix of output neuron answers of each network from the ensemble for the recognition of a PMN-1 mine at a 10 cm distance for (**a**) SNR = 15 dB, (**b**) SNR = 10 dB, (**c**) SNR = 5 dB.

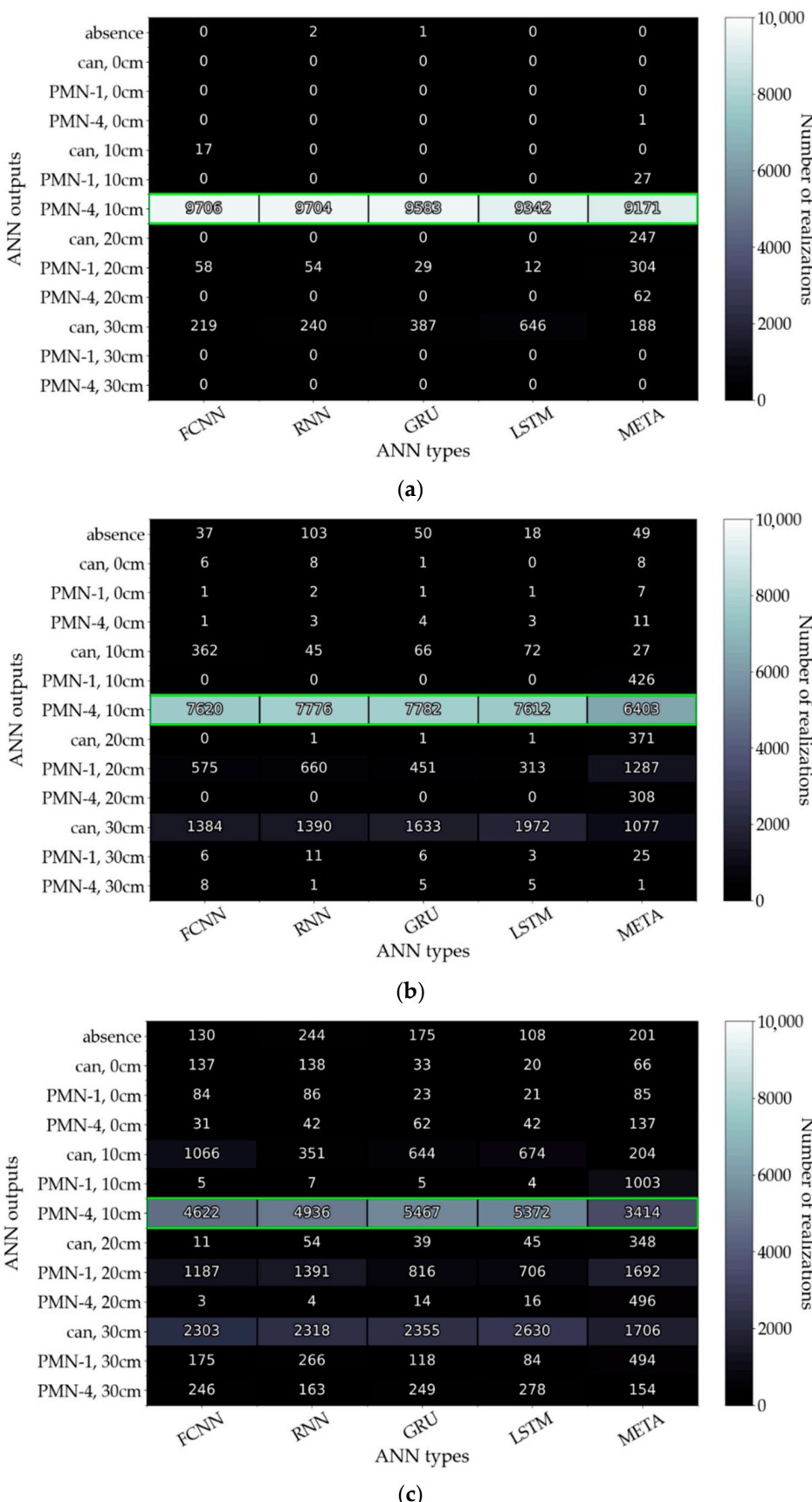

**Figure A8.** Matrix of output neuron answers of each network from the ensemble for the recognition of a PMN-4 mine at a 10 cm distance for (**a**) SNR = 15 dB, (**b**) SNR = 10 dB, (**c**) SNR = 5 dB.

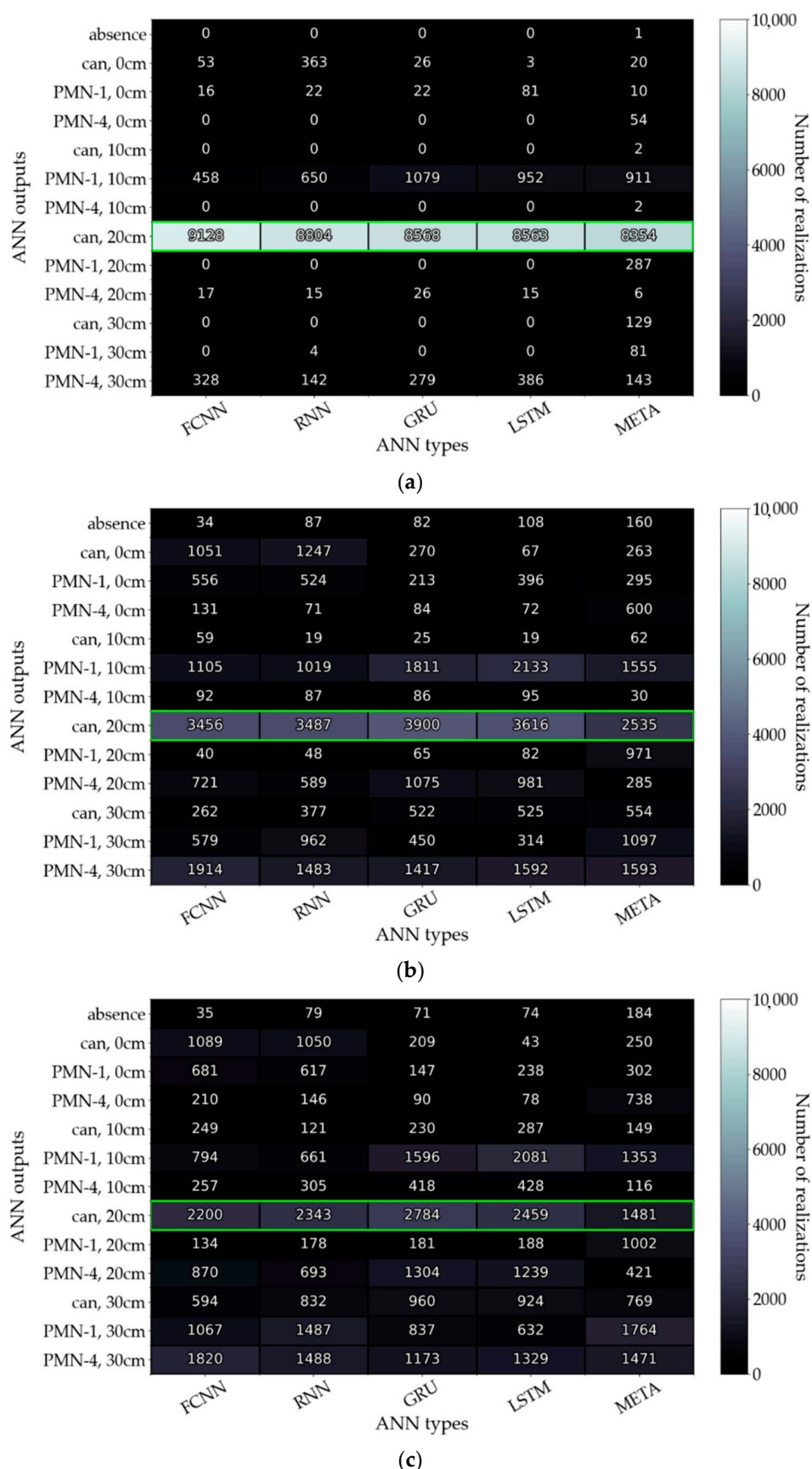

**Figure A9.** Matrix of output neuron answers of each network from the ensemble for the recognition of a metal can at a 20 cm distance for (**a**) SNR = 15 dB, (**b**) SNR = 5 dB, (**c**) SNR = 0 dB.

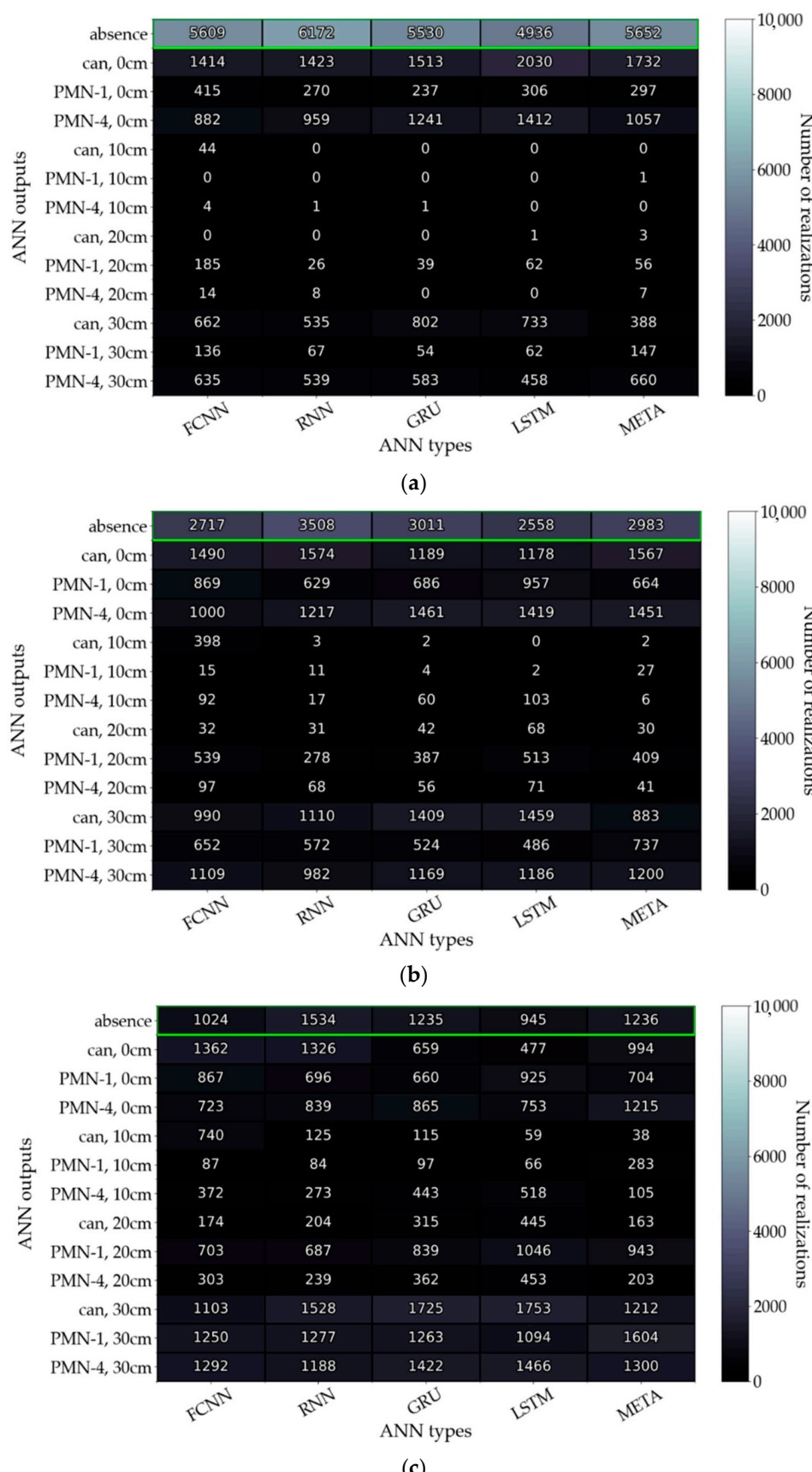

**Figure A10.** Matrix of output neuron answers of each network from the ensemble for the recognition of an absent object for (**a**) SNR = 20 dB, (**b**) SNR = 15 dB, (**c**) SNR = 10 dB.

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
