# Peer review of "Implementation of an Artificial Intelligence Approach to GPR Systems for Landmine Detection"

_remotesensing, doi:10.3390/rs14174421_

Round 1
Reviewer 1 Report
The authors proposed a subsurface survey approach based on AI and GPR. In this approach, ANN and ensemble learning were used to accomplish the classification task. They considerd target recognition in different directions and at different distances, respectively. Meanwhile, the effect of noise on the proposed method was also verified. The experimental results showed that AI has a promising future in the field of subsurface exploration. However, some contents need to be improved before this paper can be published.
1. The introduction contains too little content. Currently, artificial intelligence is a hot research topic. There are many relevant studies. It is recommended to add some references, such as:
- Manan, A., Kamal, K., Abro, A. G., Abdul Hussain Ratlamwala, T., Fahad Sheikh, M., & Zafar, T. (2021). Failure Prediction and Classification in Natural Gas Pipe-Lines Using Artificial Intelligence. Available at SSRN 3872433.
- Gu, J., Wu, L., Chen, J., Cai, R., & Wan, H. Intelligent monitoring of subsidence cracks in underground power utility tunnel. In Proc. of SPIE Vol (Vol. 12166, pp. 121663D-1).
- Zhang, Y., & Yuen, K. V. (2021). Crack detection using fusion features‐based broad learning system and image processing. Computer‐Aided Civil and Infrastructure Engineering, 36(12), 1568-1584.
2. Most of the images in the manuscript are very unclear. It is recommended that a clear version of each image be provided.
3. In section 3.4, the authors proposed an ensemble learning strategy. As seen in Figures 31-35, it seems that RNN has high recognition accuracy? It is recommended to list the recognition accuracy of each algorithm and the ensemble model in this section.
4. For classification issues, there are many evaluation metrics. It is suggested that the authors use multiple metrics to evaluate the classification effectiveness of the model.
5. The authors used the addition of noise to increase the sample numbers. As the authors say, this is a very classical approach to data augmentation. The rapid development of artificial intelligence has also contributed to the development of data augmentation methods. It is suggested that the authors could use artificial intelligence methods to improve the sample numbers in the future.
Reviewer 2 Report
I have put all the comments in the DOC document and in the PDF manuscript.
You will se that I have stopped the review in the 3.3 section because I am missing information from the authors in order to continue.

Author Response
Hello, Gennadiy Pochanin has provided me with a document that will hopefully provide some information for Reviewer 2 in order to continue the review. His comment is below, and I have attached the file he has sent to be uploaded.
Thanks,
Froney Crawford
"Reviewer 2 has stopped the review because he/she is missing information from the authors in order to continue. I attached the file with explanations and hope that respected Reviewer 2 will receive the necessary data to continue the review process."

Reviewer 3 Report
The authors present an ANN approach for landmine recognition and demonstrate its robustness against noise contamination with simulation and field experiments. However, there are some problems to be addressed:
- Originality problem. Plenty of ANN-based approaches are already been proposed for subsurface target recognition. However, this manuscript dose not demonstrate the superiority of this approach over the state-of-the-art ANN (or traditional) approaches. Please focus on the benefits and advantages of the proposed approach and provide sufficient evidence to justify the contribution of this manuscript to the GPR community.
- Article organization problems. This manuscript are overly long with redundancies and lack of focus. (1)In Abstract, we expect to find motivation of the idea, aim as well as description of the approach, and most importantly, the quantitative improvements. However, the Abstract details the radar system and the prepossessing procedure which can be discussed in following sections to avoid distracting the readers’ (2) In Introduction, the literature review paragraphs list the existing approaches without discussion of their advantages and shortcomings. Please add detail review about the state-of-the-art ANN approaches so we can follow the authors’ thought and find out what problem dose the proposed approach intend to solve. In addition, the description of the novelty is not clear. (3) In Statement of the Problem section, the manuscript details the structures of different mines but dose not explain how do the differences effect the radar signal and how can we make use of these differences to solve the target recognition problem. Please simplify the description of irrelevant factors. (4) Please highlight the contribution of this work in the Conclusion section by quantitative metrics.
- Need to improve the understanding of the background and physical process. The proposed approach mainly deals with noise contamination and low SNR signals. However, as most of antipersonnel mines are buried in shallow depth, the target echo is usually stronger than noise but much weaker than clutters such as direct coupling, ground reflection and multiple reflections between antennas and ground. These clutters often overlap with the target echo in time domain and degrade the signal-to-clutter ratio, which hinder the target detection and recognition. Please add description of signal model and figure out how to coupe with overlapped clutters.
- Make the description of the approach more clear. Please add a flow chart of the work pipeline and a figure explaining the structure of the ANN.
- Improve the presentation of the results. (1) The resolution of many figures and tables (such as Figures 11, 16 and 31 ) are not high enough. (2) Please clarify what dose the bright lines stand for in Figures 18 though 23. The detection rate? (3) What are the detection rates and false alarm rates of all these experiments? According to Figure 31, the miss rate seems unacceptable, especially under low SNR conditions. Please compare the detection performance of the proposed approach with that of other approaches. Do the proposed approach outperform the state-of-the-art approaches? (4) In section 3.4, a neural network ensemble approach is introduced. Please clarify the improvement of detection performance after adopting this approach.
Round 2
Reviewer 2 Report
Dear Authors,
The first 3 sheets of the attached pdf are my comments to this second revision of your paper and then there is the manuscript with further comments.
You have done a very interesting and very extensive work at the computational and experimental level. But my opinion is that some theoretical principles in GPR multisignal processing are missing.
My comments go in this direction and it would be good if you could make the indicated corrections. These are quick modifications that are feasible to make in a short time.
I send you my best regards and best wishes for success.
Translated with www.DeepL.com/Translator (free version)

Author Response
Please see the attached reply, along with an appended draft which is color coded for reference (both in a single document).

Reviewer 3 Report
no more comment
Author Response
Are there more changes to be made? I think the second report said no, but it is asking for a reply.